# Deletion of IFT20 exclusively in the RPE ablates primary cilia and leads to retinal degeneration

Viola Kretschmer[1☺], Sandra Schneider[1☺], Peter Andreas Matthiessen[1‡], Dominik Reichert[1,2‡], Nathan Hotaling[3], Gunnar Glaßer[4], Ingo Lieberwirth[4], Kapil Bharti[2], Rossella De Cegli[5], Ivan Conte[5,6], Emeline F. Nandrot[7], Helen Louise May-Simera[1] *

1 Faculty of Biology, Institute of Molecular Physiology, Johannes Gutenberg-University, Mainz, Germany, 2 National Eye Institute, National Institutes of Health, Bethesda, Maryland, United States of America, 3 National Center for Advancing Translational Sciences, National Institutes of Health, Bethesda, Maryland, United States of America, 4 Max Planck Institute for Polymer Research, Mainz, Germany, 5 Telethon Institute of Genetics and Medicine (TIGEM), Pozzuoli, Italy, 6 University of Naples "Federico II", Naples, Italy, 7 Sorbonne Université, INSERM, CNRS, Institut de la Vision, Paris, France

☺ These authors contributed equally to this work.
‡ PAM and DR also contributed equally to this work.
* may-simera@uni-mainz.de

**Data Availability Statement:** All relevant data are within the paper and its Supporting Information files. The whole set of transcriptomic results is available in the GEO database (GSE144724). The

## Abstract

Vision impairment places a serious burden on the aging society, affecting the lives of millions of people. Many retinal diseases are of genetic origin, of which over 50% are due to mutations in cilia-associated genes. Most research on retinal degeneration has focused on the ciliated photoreceptor cells of the retina. However, the contribution of primary cilia in other ocular cell types has largely been ignored. The retinal pigment epithelium (RPE) is a monolayer epithelium at the back of the eye intricately associated with photoreceptors and essential for visual function. It is already known that primary cilia in the RPE are critical for its development and maturation; however, it remains unclear whether this affects RPE function and retinal tissue homeostasis. We generated a conditional knockout mouse model, in which IFT20 is exclusively deleted in the RPE, ablating primary cilia. This leads to defective RPE function, followed by photoreceptor degeneration and, ultimately, vision impairment. Transcriptomic analysis offers insights into mechanisms underlying pathogenic changes, which include transcripts related to epithelial homeostasis, the visual cycle, and phagocytosis. Due to the loss of cilia exclusively in the RPE, this mouse model enables us to tease out the functional role of RPE cilia and their contribution to retinal degeneration, providing a powerful tool for basic and translational research in syndromic and non-syndromic retinal degeneration. Non-ciliary mechanisms of IFT20 in the RPE may also contribute to pathogenesis and cannot be excluded, especially considering the increasing evidence of non-ciliary functions of ciliary proteins.

title of the series is "Transcriptome profile of IFT20 Floxed mouse RPE cells all positve for DCT Cre".

**Funding:** The authors would also like to acknowledge their funding sources, the Alexander von Humboldt Foundation (Sofja Kovalevskaya Award to HMS), the Deutsche Forschungsgemeinschaft SPP2127 (DFG Grant MA 6139/3-1 and MA 6139/5-1 to HMS) and the Studienstiftung des Deutschen Volkes (to PM). The funders had no role in study design, data collection and analysis, decision to publish, or preparation of the manuscript.

**Competing interests:** The authors have declared that no competing interests exist.

**Abbreviations:** AMD, age-related macular degeneration; AR, aspect ratio; BP, biological process; DC-ERG, direct-coupled electroretinography; DEG, differentially expressed gene; DIC, differential interference contrast; ERG, electroretinogram; FO, fast oscillation; GOEA, gene ontology enrichment analysis; HBSS, Hank's balanced salt solution; IFT, intraflagellar transport; LP, light peak; OCT, optical coherence tomography; OMR, optomotoric reflex; POS, photoreceptor outer segment; RP, retinitis pigmentosa; RPE, retinal pigment epithelium; SEM, scanning electron microscopy; TEM, transmission electron microscopy.

## Introduction

Vision loss affects more than 200 million people globally, and with an aging population, it is set to increase exponentially. A multitude of retinal diseases, responsible for vision loss, are genetically inherited, with 300 retinal disease-causing genes identified so far. Strikingly, at least 50% of these genes encode cilia-associated proteins. Primary cilia are microtubule-based sensory organelles found on almost every eukaryotic cell type and are crucial for many developmental and physiological processes. They are comprised of an axoneme, 9 microtubule doublets extending from the basal body, and a ciliary membrane, which is continuous with the cell membrane. Due to an accumulation of receptors in the ciliary membrane, primary cilia coordinate several signaling pathways such as Wnt, Hedgehog (Hh), and transforming growth factor-β (Tgf-β), all indispensable for cell differentiation, organogenesis, and tissue homeostasis [1–5].

Defects in primary cilia function or assembly are responsible for a wide range of diseases, collectively termed ciliopathies. Retinal degeneration is one of the most common phenotypes associated with all syndromic ciliopathies. Furthermore, primary cilia dysfunction has been associated with a growing number of non-syndromic retinal dystrophies, such as retinitis pigmentosa (RP) [2,6]. In the vertebrate eye, primary cilia are present in a variety of different cell types [2]. So far, most research on retinal degeneration in ciliopathies has predominantly focused on the highly specialized primary cilium of the retinal photoreceptor cell, which elaborates in a connecting cilium followed by a sensory photoreceptor outer segment (POS), constituted of stacked membranous discs enriched for components of the phototransduction machinery. However, the contribution of defective primary cilia in other ocular cell types has not yet been comprehensively studied [2,7].

We have recently shown that primary cilia in the retinal pigment epithelium (RPE) are crucial for its development and maturation. Furthermore, ciliary defects impair RPE functions that are essential for photoreceptor health and activity [7,8]. The RPE is a monolayer of pigmented epithelial cells located between the neural retina and the choriocapillaris. With their long apical microvilli, RPE cells closely interact with photoreceptors in order to maintain visual function and engulf the light-sensitive POS [9–11]. Due to its diverse roles, the RPE is indispensable for health, maintenance, and function of the photoreceptor cells and, therefore, for vision. RPE dysfunction can lead to retinal degeneration and blindness, and has been associated with inherited rod-cone dystrophies and age-related macular degeneration (AMD), the most common cause of irreversible blindness in the elderly population [10–13].

Whether ciliary dysfunction in the RPE disrupts RPE function, and how this might contribute to vision loss or retinal degeneration, is not known. Therefore, we sought to generate a conditional targeted ciliopathy in the RPE. To do this we analyzed a conditional *Intraflagellar transport protein 20 homolog* (*Ift20*) knockout mouse model, in which primary cilia were exclusively ablated in the RPE [14,15]. The *Ift20* gene encodes an intraflagellar transport (IFT) protein essential for assembly, maintenance, and function of primary cilia. Defects in IFT have been shown to impair ciliogenesis [2,16,17] and loss of IFT20 is seen as a good model for loss of ciliary function.

Our data provide evidence that loss of primary cilia exclusively in the RPE results in physiological defects affecting RPE homeostasis and function and demonstrate that disruption of this process may contribute to aberrant retinal abnormalities. For the first time, we could show that ciliary defects in the RPE can precede retinal degeneration and subsequent visual impairment.

## Results

### Deletion of *Ift20* in the RPE ablates primary cilia in the RPE without affecting retinal development

In order to abolish primary cilia exclusively in the RPE, conditional *Ift20* knockout mice (*Ift20-null;Tyrp2-Cre*) were generated by crossing *Ift20^flox* mice with a *Tyrosinase-related protein-2 (Tyrp2)-Cre* transgenic mouse line, in which Cre activity in the eye was observed from mouse embryonic day E9.5 onwards [15]. Consistent with the gradual *Tyrp2* expression in the developing RPE at E11.5, Cre activity was observed mainly in the dorsal RPE with "patchy" activity in the ventral RPE. At E13.5, most of the RPE cells showed Cre activity [14,15]. This results in a remediated deletion of exons 2 and 3, leading to the loss of the start codon and thus producing a null allele for *Ift20* [18]. Ablation of *Ift20* eliminates the complete ciliary structure since IFT20 assists in the transport of ciliary membrane proteins from the Golgi complex to the cilium [17], essential for assembly, maintenance and function of primary cilia [16,17].

We stained RPE flatmounts of E16.5-old mice for IFT20 and the *cis*-Golgi matrix protein GM130 to assess recombination efficiency (**S1A Fig**) [17]. Quantifying only GM130 positive cells, we found that *Ift20^null;Tyrp2-Cre* mice showed a significant reduction in IFT20 expression in the RPE (5.3%) compared to controls (98.5%) (**S1B Fig**). To determine whether this reduction of IFT20 expression in the RPE leads to a loss of cilia, we identified primary cilia via co-localization of ciliary membrane marker ARL13B and transition zone marker GT335 at E16.5, when RPE primary cilia are longest and most abundant (**Fig 1A**) [19]. Quantification confirmed a significant decrease in ciliation in *Ift20^null;Tyrp2-Cre* RPE (total ciliation = 12.5%) compared to controls (total ciliation = 84.3%) (**Fig 1B**). The few cilia remaining in the mutant appeared longer, suggesting a defect in ciliary trafficking and length regulation (**Fig 1C**).

Despite loss of primary cilia in the RPE, adjacent photoreceptor primary cilia and the retina developed normally. Histological sections and transmission electron microscopy (TEM) of eye sections from 1-month-old mice showed that deletion of *Ift20* specifically in the RPE does not affect the development and maturation of the neural retina (**Fig 1D**). Furthermore, in *Ift20^null; Tyrp2-Cre* mice we were able to observe fully matured photoreceptors, exhibiting a typical photoreceptor primary cilium with a basal body, transition zone (connecting cilium) and axoneme, extending into the outer segment (**Fig 1E and 1F**). To confirm that our Cre driver activity was restricted to the RPE, we crossed the *Tyrp2-Cre* line with a tdTomato reporter mouse, namely Ai14. This mouse line contains a floxed stop cassette upstream of the gene for the red fluorescent protein tdTomato [20] and only expresses tdTomato in cells with Cre activity. Fluorescent analysis of ocular sections confirmed that *Tyrp2-Cre* activity was only found in the RPE (**S1C Fig**).

### IFT20 loss causes subtle alterations in RPE ultrastructure

We analyzed the ultrastructure of the RPE and adjacent POS in 1- and 4-months-old mice via TEM. In the mutant at 1 month of age, we observed an increased number of "subretinal gaps" (**Fig 1G**). Despite these structures occasionally being found in the control, they were more commonly observed in *Ift20^null;Tyrp2-Cre* RPE (**S2A Fig**) and had never been seen in preparations from ciliopathy mutant mice before.

At 4 months of age these "gaps" were no longer present, instead we observed numerous abnormalities in mutant RPE cells. These include regions of multilayered cells (**Fig 1H, top right panel**) and abnormal accumulation of material, possibly detached microvilli or unphagocytosed POS (dashed outline) that are not in connection with the apical RPE (red arrow), in the subretinal space (**Fig 1H, bottom right panel**). Despite seeing changes of the

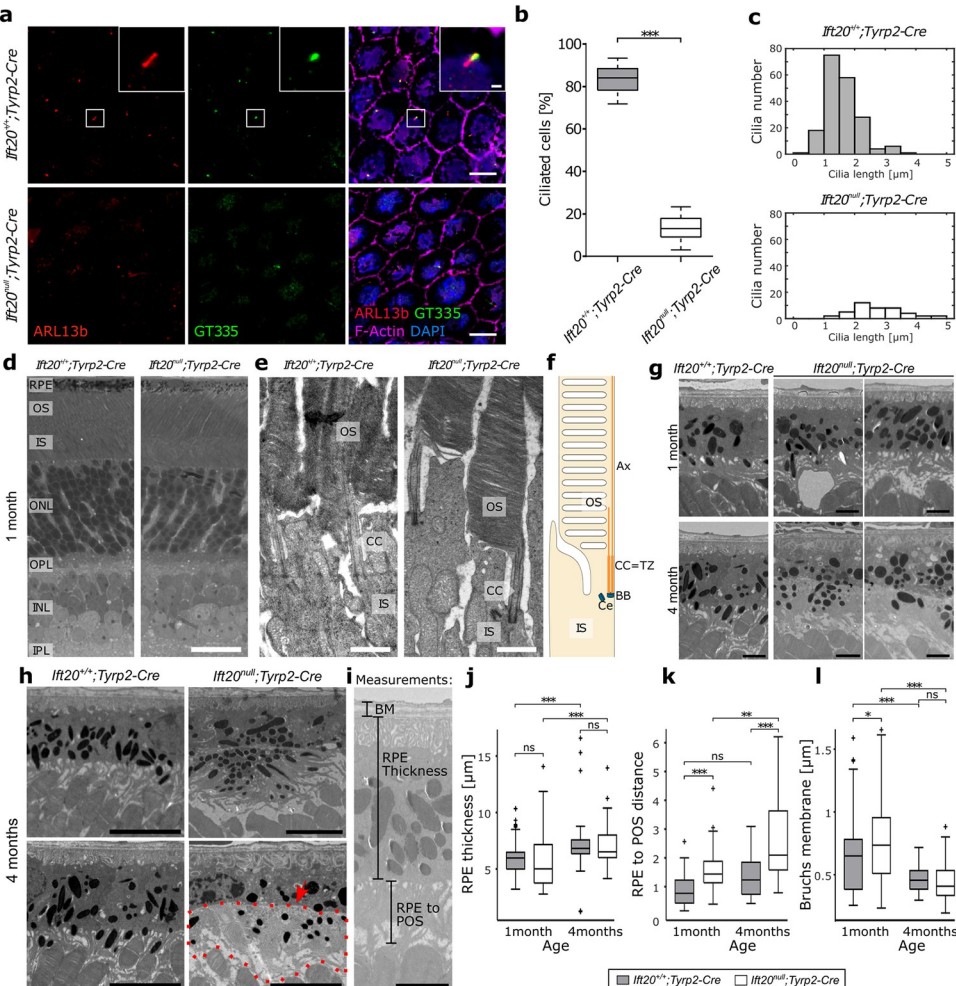

**Fig 1. Conditional knockout of *Ift20* ablates primary cilia in the RPE without affecting other retinal layers and causes subtle alterations in RPE ultrastructure.** (**a**) Representative fluorescent images of E16.5 RPE flatmounts stained for ARL13B (red) and GT335 (green) to visualize primary cilia. Staining for F-Actin (magenta) was used to visualize the cytoskeleton, DAPI to stain nuclear DNA. Scale bars: 10 μm. (**b**) Quantification of primary cilia in E16.5 RPE revealed that *Ift20^{null}*;*Tyrp2-Cre* RPE showed significantly less ciliated cells (13% *n* = 4 eyes (1,039 cells)) compared to controls (84.3% *n* = 4 eyes (816 cells)). Statistical analysis was performed using ROUT test (Q = 0.1%) before using unpaired *t* test ($p < 0.001$). Median: *Ift20^{+/+}*;*Tyrp2-Cre* 84.75% *Ift20^{null}*;*Tyrp2-Cre* 13.21%. (**c**) Cilia length data binned at intervals of 0.5 μm. Cilia remaining in the mutant where more likely to be longer than in control. *Ift20^{+/+}*;*Tyrp2-Cre* (*n* = 4 eyes 191 cilia, *Ift20^{null}*;*Tyrp2-Cre* *n* = 4 eyes 43 cilia). (**d**) Representative images of histological eye sections from 1-month-old mice. In both *Ift20^{null}*;*Tyrp2-Cre* and control RPE, no differences in retinal layers could be observed. Scale bar: 25 μm. (**e**) Representative TEM image of a photoreceptor connecting cilium from control and mutant mice. Photoreceptor connecting cilia remained intact. Scale bar: 1 μm. (**f**) Schematic showing a typical photoreceptor primary cilium. (**g**) Representative TEM images of eye sections showing subretinal gaps in *Ift20^{null}*; *Tyrp2-Cre* mice at 1 month of age, while they appear to be gone by 4 months. Scale bars: 10 μm. (**h**) Representative TEM images of eye sections from 4-month-old mice showing multilayered RPE cells (top), as well as the absence of microvilli and accumulation of debris (bottom, red outline, arrow highlights the continuous apical membrane) in *Ift20^{null}*;*Tyrp2-Cre* mice. Scale bars: 10 μm. (**i**) Measurements taken from TEM images: Bruch's membrane, RPE thickness and RPE to POS distance. (**j**) Quantification of RPE thickness measured in *Ift20^{+/+}*;*Tyrp2-Cre* and *Ift20^{null}*; *Tyrp2-Cre* revealed significant differences between 1 and 4 months, but no differences between control and knockout. Median: 1 month *Ift20^{+/+}*;*Tyrp2-Cre* 6.0 μm, *Ift20^{null}*;*Tyrp2-Cre* 5.0 μm. 4 months: *Ift20^{+/+}*;*Tyrp2-Cre* 6.8 μm, *Ift20^{null}*; *Tyrp2-Cre* 6.5 μm. (**k**) Quantification of RPE to POS distance measured in *Ift20^{+/+}*;*Tyrp2-Cre* and *Ift20^{null}*;*Tyrp2-Cre* shows highly significant differences between control and knockout. Median: 1 month *Ift20^{+/+}*;*Tyrp2-Cre* 0.77 μm, *Ift20^{null}*;*Tyrp2-Cre* 1.38 μm. 4 months: *Ift20^{+/+}*;*Tyrp2-Cre* 1.24 μm, *Ift20^{null}*;*Tyrp2-Cre* 2.09 μm. (**l**) Quantification of Bruch's membrane thickness measured in *Ift20^{+/+}*;*Tyrp2-Cre* and *Ift20^{null}*;*Tyrp2-Cre* reveals only slight increase in the mutant at 1 month of age. Median: 1 month *Ift20^{+/+}*;*Tyrp2-Cre* 0.65 μm, *Ift20^{null}*;*Tyrp2-Cre* 0.75 μm. 4 months: *Ift20^{+/+}*;*Tyrp2-Cre* 0.46 μm, *Ift20^{null}*;*Tyrp2-Cre* 0.48 μm. Statistical analysis was performed using an unpaired two-tailed *t* test. (For j–l: 1 month *Ift20^{+/+}*;*Tyrp2-Cre* *n* = 6 eyes, >60 measurements, *Ift20^{null}*;*Tyrp2-Cre* *n* = 6 eyes, >60 measurements;

4 months $Ift20^{+/+}$;$Tyrp2$-$Cre$ n = 4 eyes, >50 measurements, $Ift20^{null}$;$Tyrp2$-$Cre$ n = 4 eyes, >50 measurements). Significance levels: >0.05 not significant (ns), <0.05 *, <0.01 **, <0.001 ***. Box plots: Box limits represent the first and third quartile, the central line shows the median and the whiskers indicate the 5th and 95th percentile. RPE, retinal pigment epithelium; OS, outer segments; IS, inner segments; ONL, outer nuclear layer; OPL, outer plexiform layer, INL, inner nuclear layer; IPL, inner plexiform layer; Ce, centriole; CC, connecting cilium; Ax, axoneme; TZ, transition zone; BB, basal body; BM: Bruch's membrane; POS, photoreceptor outer segment; TEM, transmission electron microscopy. Numerical data can be found in S6 Table.

microvilli structures upon TEM, scanning electron microscopy (SEM) at an earlier time point (S2B Fig) did not show any differences. However, these studies could only be done at P0, since the close association between the RPE and POS prohibits a clean separation between the 2 tissues at later ages.

In an attempt to quantify these ultrastructural changes, we measured the thickness of the RPE, the width of the Bruch's membrane, as well as the distance between the apical RPE membrane and end of the POS (RPE to POS distance) (Fig 1I). Although there was no significant difference in RPE thickness between control and mutant at either of the 2 time points measured (Fig 1J), the RPE to POS distance was significantly increased at both stages (Fig 1K). The thickness of the Bruch's membrane was only slightly increased in the mutant at 1 month of age (Fig 1L). We also noticed that the centered elastin layer of the Bruch's membrane became increasingly discontinuous in the mutant (red brackets) (S2D Fig).

TEM also revealed RPE cells from $Ift20^{null}$;$Tyrp2$-$Cre$ mice displaying abnormal pigmentation (S2E Fig). Examples include 2 "normal" RPE cells flanking a cell almost completely devoid of melanosomes at 1 month of age, as well as an RPE cell devoid of melanosomes in the cell body, flanked by an RPE cell with excessive accumulation of melanosomes at 4 months of age. However, when quantifying melanosomes in flatmount preparations, no significant variation was observed in the overall melanosome content (S2F and S2G Fig).

## Single-cell-resolution analysis of RPE flatmounts revealed only minor changes to cellular morphology

To further probe the phenotype of $Ift20^{null}$;$Tyrp2$-$Cre$ mice, we examined epithelial patterning at P11, P29 and 3 months of age. At P11, the RPE is considered mature [21,22], yet the retinal photoreceptors are not completely developed. By P29, the photoreceptors have fully formed and are ensheathed by the apical processes of the RPE, forming a functional unit [23–25]. By 3 months of age, the adult RPE has been fully functional for several weeks. We assessed changes in cellular morphology via high-content image analysis of entire RPE flatmounts from both $Ift20^{+/+}$;$Tyrp2$-$Cre$ and $Ift20^{null}$;$Tyrp2$-$Cre$ mice stained with phalloidin. Cell borders were recognized and segmented using REShAPE, a machine learning based segmentation tool for RPE monolayer [26]. The resultant images outlining the cell borders were used to calculate cell morphometry for individual RPE cells (Fig 2A). We analyzed 4 morphometric features, namely cell area, aspect ratio (AR), number of neighbors, and hexagonality, which have previously been shown to define the compact packing of RPE cells and change in degenerating RPE cells [26].

Upon analysis of all cells across the whole flatmount, we observed a modest increase in cell size in the mutant starting from P29 onwards (Fig 2B–2D). Binning the cell area data at intervals of 50 μm$^2$ (Fig 2C) revealed remarkably similar profiles between the control and the mutant across all 3 ages examined. Thus, suggesting no major differences in cell size ratios, despite a slight increase in total cell size in the older categories. The minor increase in cell area was accompanied by a small but significant decrease in the AR at the same ages, which is a measure of cell elongation (Fig 2E). The number of neighbors and hexagonality score was not significantly different at any stage (Fig 2F and 2G).

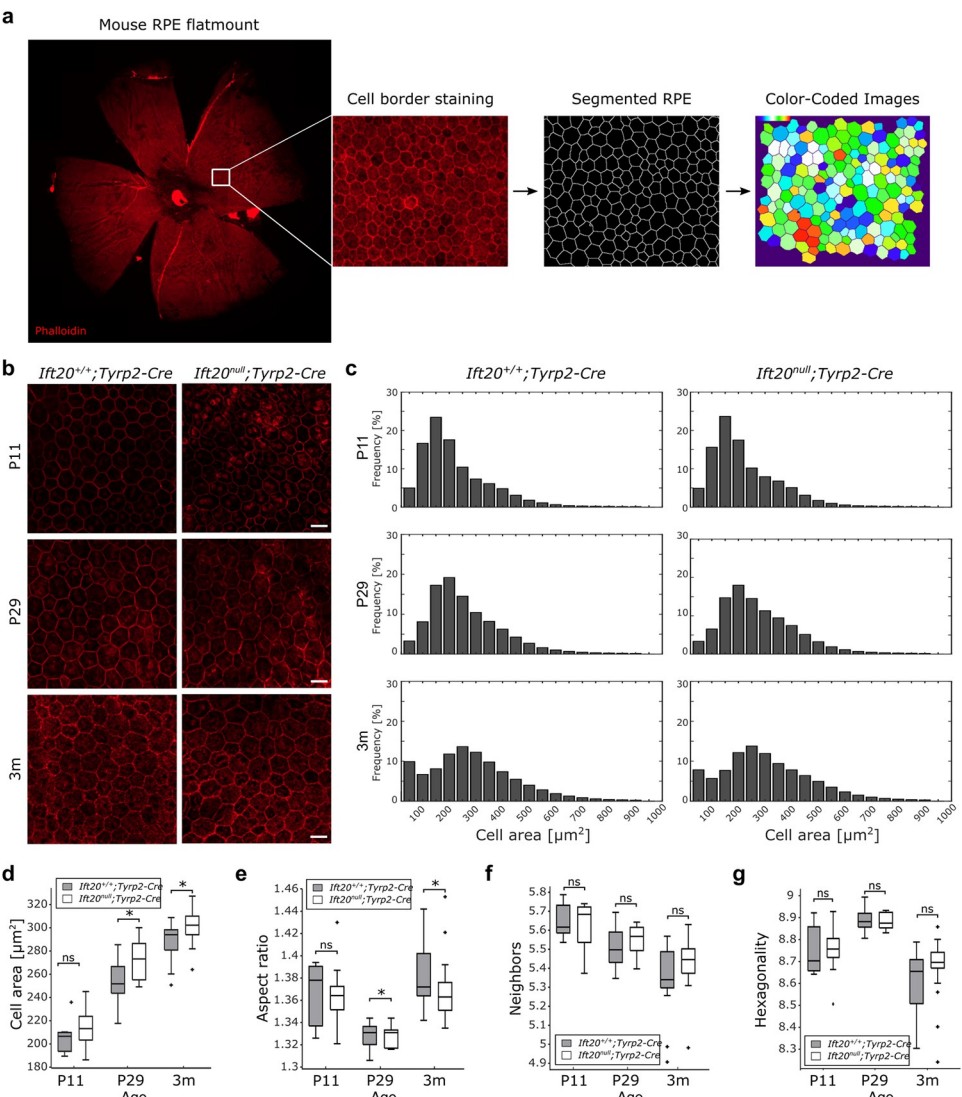

**Fig 2. Single-cell-resolution analysis of RPE flatmounts revealed only minor changes to cellular morphology.** (**a**) *Ift20^{+/+};Tyrp2-Cre* and *Ift20^{null};Tyrp2-Cre* RPE flatmounts stained with Phalloidin (red), cell borders were automatically analyzed, segmented, and visualized by color-coded images. (**b**) Representative images showing Phalloidin staining (red) at P11, P29, and 3M of *Ift20^{+/+};Tyrp2-Cre* and *Ift20^{null};Tyrp2-Cre* RPE flatmounts. Scale bar: 20 μm. (**c**) Cell area data binned at intervals of 50 μm². Binned profiles look remarkably similar between control and mutant across all 3 ages examined. (**d–g**) Quantification of the cell size, AR, number of neighbors, and hexagonality of *Ift20^{+/+};Tyrp2-Cre* and *Ift20^{null};Tyrp2-Cre* RPE flatmounts. Analysis of cell size revealed a modest increase in cell size in the mutant starting from P29 onwards, while AR was decreasing. No changes were detected in number of neighbors and hexagonality. Median: cell area P11 *Ift20^{+/+};Tyrp2-Cre* 208.67 μm², *Ift20^{null};Tyrp2-Cre* 218.08 μm². P29: *Ift20^{+/+};Tyrp2-Cre* 249.7 μm², *Ift20^{null};Tyrp2-Cre* 271.84 μm². 3m: *Ift20^{+/+};Tyrp2-Cre* 287.82 μm², *Ift20^{null};Tyrp2-Cre* 303.72 μm². Median: aspect ratio P11 *Ift20^{+/+};Tyrp2-Cre* 1.375, *Ift20^{null};Tyrp2-Cre* 1.365. P29: *Ift20^{+/+};Tyrp2-Cre* 1.331, *Ift20^{null};Tyrp2-Cre* 1.335. 3m: *Ift20^{+/+};Tyrp2-Cre* 1.375, *Ift20^{null};Tyrp2-Cre* 1.363. Median: neighbors P11 *Ift20^{+/+};Tyrp2-Cre* 5.61, *Ift20^{null};Tyrp2-Cre* 5.68. P29: *Ift20^{+/+};Tyrp2-Cre* 5.51, *Ift20^{null};Tyrp2-Cre* 5.59. 3m: *Ift20^{+/+};Tyrp2-Cre* 5.34, *Ift20^{null};Tyrp2-Cre* 5.43. Median: hexagonality P11 *Ift20^{+/+};Tyrp2-Cre* 8.70, *Ift20^{null};Tyrp2-Cre* 8.73. P29: *Ift20^{+/+};Tyrp2-Cre* 8.88, *Ift20^{null};Tyrp2-Cre* 8.86. 3m: *Ift20^{+/+};Tyrp2-Cre* 8.60, *Ift20^{null};Tyrp2-Cre* 8.64 μm². n numbers P11: *Ift20^{+/+};Tyrp2-Cre* (7 flat mounts, 163,530 cells); *Ift20^{null};Tyrp2-Cre* (14 flat mounts 295,059 cells), P29: *Ift20^{+/+};Tyrp2-Cre* (11 flat mounts, 274,428 cells); *Ift20^{null};Tyrp2-Cre* (9 flat mounts, 227,064 cells), 3m: *Ift20^{+/+};Tyrp2-Cre* (13 flat mounts, 510,473 cells); *Ift20^{null};Tyrp2-Cre* (18 flat mounts 675,748 cells). Significance levels: >0.05 not significant (ns), <0.05 *, <0.01 **, <0.001 ***. Box plots: Box limits represent the first and third quartile, the central line shows the median and the whiskers indicate the 5th and 95th percentile. Numerical data can be found in S6 Table. AR, aspect ratio; RPE, retinal pigment epithelium.

## Transcriptomic analysis reveals maturation defects leading to defective function in mutant RPE

To further investigate the effect of primary cilia ablation specifically in the RPE, we performed an unbiased RNA-seq analysis of RPE cells isolated from *Ift20^null^;Tyrp2-Cre* and *Ift20^+/+^; Tyrp2-Cre* mice at P11, P29, and 3 months of age. At all 3 time points, we observed a significant mis-regulation of the transcriptome compared to controls (**Fig 3A**). More specifically, at P11 we observed an up-regulation of 853 transcripts and a down-regulation of 724 transcripts. At P29, we observed an up-regulation of 403 transcripts and a down-regulation of 300 transcripts. At 3 months, we observed an up-regulation of 454 transcripts and a down-regulation of 145 transcripts (**Fig 3A** and **S1 Table**). We compared the differentially expressed genes (DEGs) with a list of RPE signature genes published in Strunnikova and colleagues [27] (**Fig 3B** and **S2 Table**). The number of differentially expressed RPE signature genes decreased as the tissue matured; however, across all ages more genes were down-regulated than up-regulated compared to controls, suggesting a less robust RPE transcriptome (**Fig 3A and 3B**).

In an attempt to further determine the biological consequences of these differentially expressed transcripts, we performed a gene ontology enrichment analysis (GOEA), restricting the output to biological process (BP) terms at each time point (**S3A and S3B Fig** and **S3 Table**). At P11, many of the down-regulated transcripts correlated with BPs associated with development, differentiation, and maturation. These include "epithelial cell proliferation," "regulation of signal transduction," and "regulation of multicellular organismal development." At P29, we found more changes in transcripts related to RPE function, with BPs including "visual perception," "ion transmembrane transport," and "response to cAMP." In contrast to P11, we found many more BPs up-regulated than down-regulated at 3 months of age.

Although all 3 time points reflect different stages of RPE maturation and pathogenic processes, it is still interesting to compare differentially expressed transcripts common across all 3 data points, as these might reflect possible underlying mechanisms (**S4 Table**): We identified 50 genes that were consistently up-regulated, while only 16 genes were consistently down-regulated. Consistently down-regulated genes included *Rpe65*, a key component of the visual cycle [10,11]. The most strongly inhibited gene was Nog (Noggin) (refer to **S4 Table**). *Nog* has been described as an anti-EMT (epithelial-to-mesenchymal transition) gene, as its overexpression suppresses EMT [28–30]. Moreover, the 4 most highly up-regulated genes were all keratins (*Krt15*, *Krt4*, *Krt5*, and *Krt6a*) (**S4 Table**). Mis-regulation of keratins has been shown to contribute to EMT [31]. This data is in line with our previous findings that loss of ciliary protein *Bbs8* in the RPE leads to a pathological phenotype involving EMT [8] and is consistent with the minor increase in cell size and increased AR of *Ift20^null^;Tyrp2-Cre* RPE.

Furthermore, among the common DEGs we observed significant changes in DEGs associated with epithelial characteristics (**S3B–S3D Fig** and **S5 Table**). Genes that are associated with transmembrane transport (**S3B Fig**) were down-regulated in P29 *Ift20^null^;Tyrp2-Cre* RPE, but up-regulated at 3 months of age. Furthermore, in P29 and 3-month-old *Ift20^null^;Tyrp2-Cre* RPE we found genes associated with cytoskeletal reorganization that are mis-regulated (**S3C Fig**) and genes that are associated with tissue remodeling, which were down-regulated (**S3D Fig**). These results show a switch in expression of genes associated with the epithelial characteristics of the RPE.

To examine the functional effect of primary cilia loss exclusively in the RPE, we measured the RPE response by direct-coupled ERG (dcERG) in P25 mice (**Fig 3C–3E**). In *Ift20^null^; Tyrp2-Cre* mice, we observed a significant decrease in the c-wave response ($p = 0.045$) compared to control mice. The c-wave mainly comes from subretinal $K^+$ changes, leading to $K^+$ signaling in the RPE, and reflects functional integrity of photoreceptors and RPE [32,33]. We

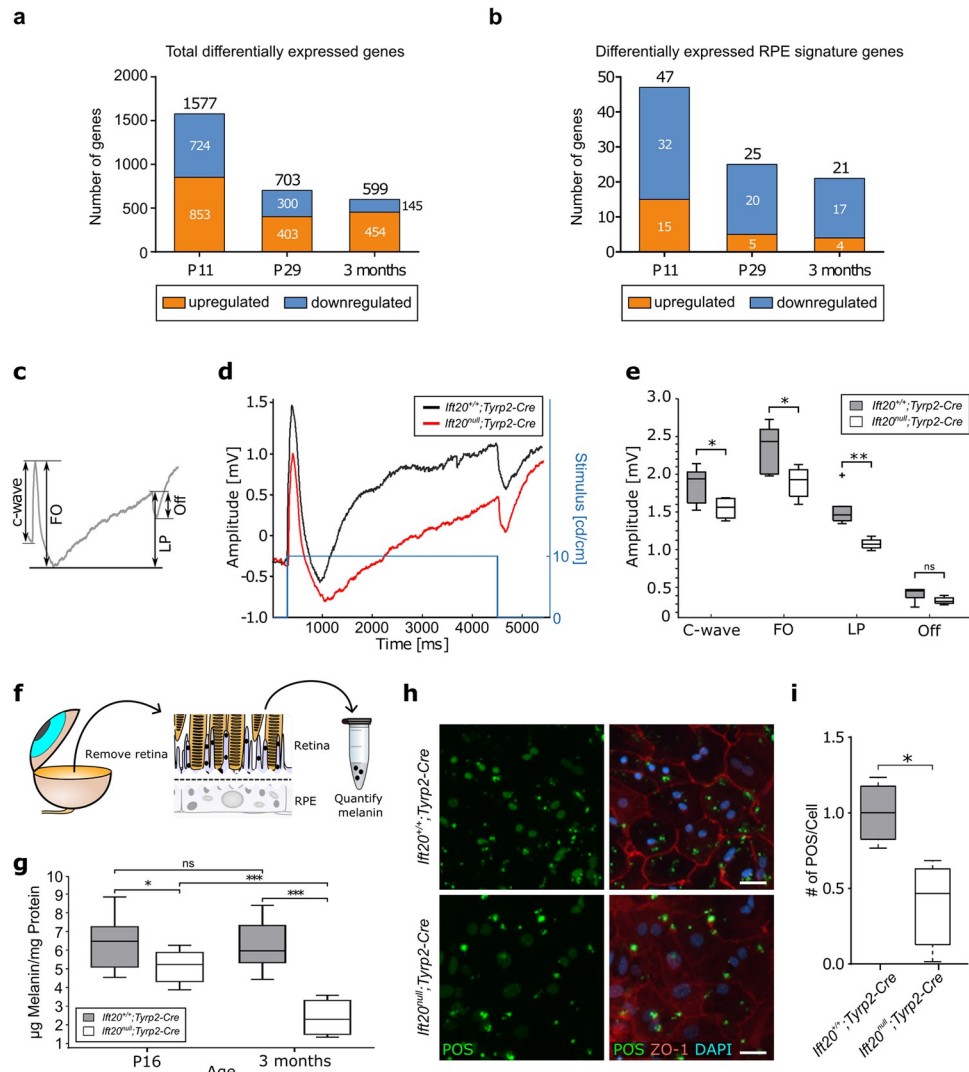

**Fig 3. Transcriptomic analysis reveals maturation defects leading to defective function in mutant RPE.** (**a**) Bar chart showing DEGs obtained via transcriptomic analysis. The number of specific and common DEGs and the orientation of expression are shown. (**b**) Bar chart depicting the number up- and down-regulated genes for RPE signature genes. At all ages, the number of differentially expressed RPE signature genes decreased; however, across all ages more genes were down-regulated than up-regulated compared to controls. (**c**) Representative DC-ERG trace showing all components of a DC-ERG response (c-wave, FO, LP, and Off). (**d**) Averaged trace of DC-ERG response from P25 *Ift20^null^;Tyrp2-Cre* mice (red) versus *Ift20^+/+^;Tyrp2-Cre* mice (black). Traces were drift-corrected and smoothed by a moving average filter (see Material and methods). Stimulus = 10 cd*s/m². (**e**) Quantification of DC-ERG responses. Significant differences were observed in c-wave ($p = 0.045$), FO ($p = 0.029$), and LP ($p = 0.0056$). The off-response showed no significance ($p = 0.15$). Median: c-wave *Ift20^+/+^;Tyrp2-Cre* 1.94 mV, *Ift20^null^;Tyrp2-Cre* 1.56 mV. FO *Ift20^+/+^;Tyrp2-Cre* 2.42 mV, *Ift20^null^;Tyrp2-Cre* 1.92 mV. LP *Ift20^+/+^;Tyrp2-Cre* 1.48 mV, *Ift20^null^;Tyrp2-Cre* 1.06 mV. Off *Ift20^+/+^;Tyrp2-Cre* 0.45 mV, *Ift20^null^;Tyrp2-Cre* 0.22 mV. Statistical analysis was performed using the unpaired two-tailed *t* test. *Ift20^+/+^;Tyrp2-Cre* n = 12 eyes, 6 mice, *Ift20^null^;Tyrp2-Cre* n = 12 eyes, 6 mice. (**f**) Schematic of experimental procedure of retinal adhesion assay. After enucleation and removal of the lens, the retina was separated from eyecup ripping off melanin containing apical microvilli. After lysis the melanin was quantified. (**g**) Quantification of melanin attached to the retina was significantly increased in P16 *Ift20^null^;Tyrp2-Cre* compared to controls ($p < 0.05$). This effect increased over time (3 months of age, $p < 0.001$). In contrast, the melanin concentration in *Ift20^+/+^;Tyrp2-Cre* retinas remained stable between both ages ($p > 0.05$). Median: P16 *Ift20^+/+^;Tyrp2-Cre* 6.47 µg melanin/mg protein, *Ift20^null^;Tyrp2-Cre* 5.22 µg melanin/mg protein; 3 months *Ift20^+/+^;Tyrp2-Cre* 5.96 µg melanin/mg protein, *Ift20^null^;Tyrp2-Cre* 2.29 µg melanin/mg protein. P16 *Ift20^+/+^;Tyrp2-Cre* (n = 12 retina); *Ift20^null^;Tyrp2-Cre* (n = 6 retina), 3 months *Ift20^+/+^;Tyrp2-Cre* (n = 6 retina); *Ift20^null^;Tyrp2-Cre* (n = 6 retina). (**f**) Representative fluorescent images of in vitro phagocytosis assay. Scale bar: 30 µm. (**h**) In vitro phagocytosis assay. Representative images of fluorescently labeled POS-fed *Ift20^+/+^;Tyrp2-Cre* and *Ift20^null^;Tyrp2-Cre* RPE cultures.

Counterstained with ZO-1 (cell borders) and DAPI (nuclei). (**i**) Quantification of POS uptake revealed a significant decrease in POS phagocytosis ($p = 0.0025$) in isolated and cultured *Ift20^null;Tyrp2-Cre* RPE primary cells compared to controls. Statistical analysis was performed using the Welch-corrected parametric unpaired *t* test ($P < 0.05$). Experiment 1: *Ift20^{+/+};Tyrp2-Cre* = 4 wells/3,396 cells, *Ift20^null;Tyrp2-Cre* = 8 wells/3,904 wells. Experiment 2: *Ift20^{+/+}; Tyrp2-Cre* = 5 wells/3,608 cells, mutant = 9 wells/4,322 cells. Experiment 3: *Ift20^{+/+};Tyrp2-Cre* = 4 wells/2,346 cells, mutant = 6 wells/2,962 cells. POS, photoreceptor outer segments; FO, fast oscillation; LP, light peak; Off, off-response. Significance levels: >0.05 not significant (ns), <0.05 *, <0.01 **, <0.001 ***. Box plots: Box limits represent the first and third quartile, the central line shows the median and the whiskers indicate the 5th and 95th percentile. Numerical data can be found in S6 Table. DC-ERG, direct-coupled electroretinography; DEG, differentially expressed gene; RPE, retinal pigment epithelium.

also observed a significant difference in fast oscillation (FO, $p = 0.029$), which suggests delayed hyperpolarization of the basolateral membrane of the RPE, probably as a delayed consequence of light-evoked subretinal $K^+$ decrease, leading to a decreased potential across the RPE (effectively causing the c-wave) (**Fig 3D and 3E**) [34]. Similarly, the light peak response (LP) was significantly reduced in *Ift20^null;Tyrp2-Cre mice* ($p = 0.0056$), indicating a possible defect in $Cl^-$ channel activity, causing depolarization of the basal membrane [35]. The off-response (Off) showed a reduced trend; however, this was not statistically significant.

RPE-photoreceptor interaction was also examined by analysis of retinal adhesion between RPE cells and adjacent photoreceptors. Normally, RPE apical microvilli form a functional connection between the RPE and POS [10,11]. Melanin synthesized by the RPE can be found in the apical processes that wrap around the POSs of the retina. This makes it a good marker for assessing the attachment of the RPE to the retina upon separation. We analyzed retinal adhesion between the POSs and the RPE in P16 and 3-month-old mice by quantifying the amount of melanin, found in RPE-derived microvilli, attached to the retina after mechanical separation from the RPE (**Fig 3F**) [36]. We observed a significantly reduced amount of melanin attached to the retina in *Ift20^null;Tyrp2-Cre* mice compared to control as early as P16, suggesting a decrease in retinal adhesion (**Fig 3G**). At 3 months, this reduction is even more pronounced. In contrast, the melanin concentration in *Ift20^{+/+};Tyrp2-Cre* retinas remained stable between these 2 time points.

To determine whether the RPE from *Ift20^null;Tyrp2-Cre* mice had a phagocytosis defect, we cultured RPE cells from *Ift20^null;Tyrp2-Cre* and *Ift20^{+/+};Tyrp2-Cre* mice and performed phagocytosis assays in vitro [37]. After being exposed to POS, we found that mutant RPE cultures were less able to phagocytose POS as compared to the control RPE (**Fig 3H and 3I**).

Combined, these results suggest that the loss of primary cilia in the RPE affected several aspects of RPE function and homeostasis. To determine whether this was enough to influence retinal health and visual function, we sought to further characterize the visual phenotype in *Ift20^null;Tyrp2-Cre* mice.

## IFT20 loss in the RPE ultimately leads to visual impairment in mice

Despite changes in RPE functionality that were observed as early as P25, retinal health, as observed via electroretinogram (ERG), and visual function, as measured via the optomotoric reflex (OMR), was initially unaffected. ERG responses were measured under both scotopic (dark-adapted, 10 cd*s/m$^2$) and photopic (light-adapted, 100 cd*s/m$^2$) conditions over a period of 12 months (**Fig 4A–4E**). Representative ERG traces of *Ift20^null;Tyrp2-Cre* and *Ift20^{+/+};Tyrp2-Cre* mice at 1 month, 6 months, and 12 months of age are shown in **Fig 4A**. In control mice, the amplitudes for both the scotopic and photopic a-waves, which reflect photoreceptor responses, and the scotopic and photopic b-waves that mirror the response of secondary neurons remained constant over a period of 12 months. At 1 month of age, only the scotopic a wave showed a slight difference between *Ift20^null;Tyrp2-Cre* mice compared to controls. At 2

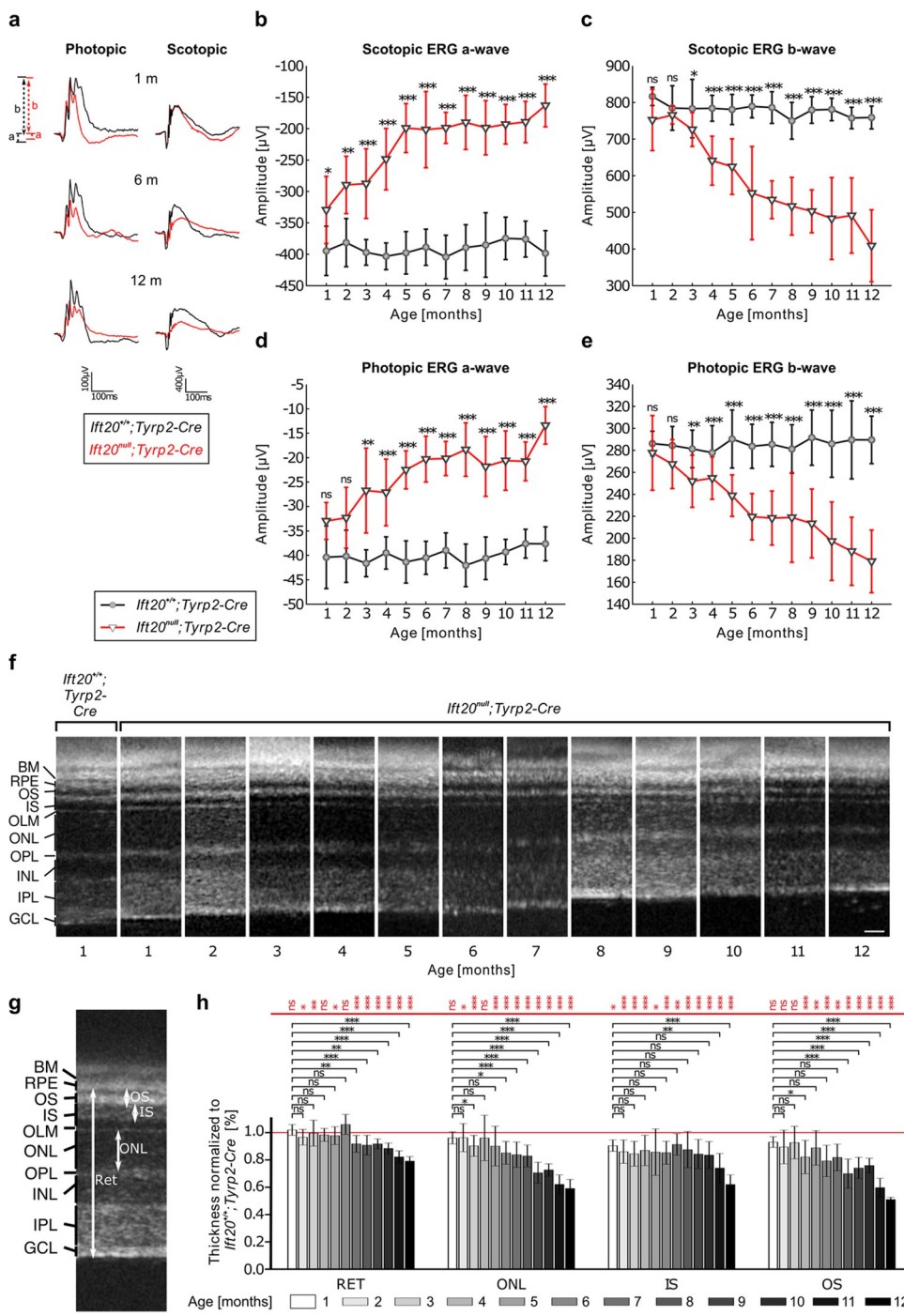

**Fig 4. Loss of IFT20 in the RPE ultimately leads to visual impairment in mice.** (**a**) Representative traces of the electric responses upon light stimulation of photoreceptors and downstream cells of *Ift20^{+/+};Tyrp2-Cre* (black) and *Ift20^{null};Tyrp2-Cre* (red) mice in photopic (100 cd*s/m²) and scotopic (10 cd*s/m²) conditions. From 1 month of age until 12 months of age, a degression of the responses was seen. (**b**) The scotopic a-wave of *Ift20^{null};Tyrp2-Cre* rods was significantly lower compared to controls. Over the course of a year, the scotopic a-wave declined to approximately 50%. (**c**) By the age of 3 months, the scotopic b-wave response of the secondary neurons was significantly lower in *Ift20^{null};Tyrp2-Cre* mice compared to controls. The scotopic b-wave also declined to approximately 50% within 1 year. (**d**) From 3 months of age, the photopic a-wave response of *Ift20^{null};Tyrp2-Cre* cones was significantly lower and declined by approximately 50% over the course of a year. (**e**) From 3 months of age, the photopic b-wave response of *Ift20^{null};Tyrp2-Cre* secondary neurons was significantly lower and declined by approximately 50% within 1 year. Statistical analysis was performed using unpaired *t* test. (**f**) Representative images of in vivo OCT scans of *Ift20^{null};*

*Tyrp2-Cre* mice from 1 to 12 months compared to 1-month-old *Ift20$^{+/+}$;Tyrp2-Cre*. Scale bar: 25 μm. (**g**) Retinal thickness (RET), as well as ONL, IS, and OS were measured 500 μm off the center of the optic nerve. (**h**) Quantification revealed a reduction in retina thickness of *Ift20$^{null}$;Tyrp2-Cre* mice over time, which was accompanied by thinning of the POSs, while inner segment thickness did not significantly decline until 10 months of age. Statistical analysis was performed using Holm–Sidak test (*Ift20$^{null}$;Tyrp2-Cre* vs. *Ift20$^{+/+}$;Tyrp2-Cre*) and Dunnet's multiple comparison (age comparison), both followed by a one-way ANOVA. BM, Bruch's membrane; RPE, retinal pigment epithelium; OS, outer segments; IS, inner segments; OLM, outer limiting membrane; ONL, outer nuclear layer; OPL, outer plexiform layer; INL, Inner nuclear layer; IPL, inner plexiform layer; GCL, ganglion cell layer. For all experiments $n > 4$ animals. Significance levels: >0.05 not significant (ns), <0.05*, <0.01**, <0.001***. Numerical data can be found in S6 Table. IFT, intraflagellar transport; OCT, optical coherence tomography; RET, retinal thickness.

months of age, this amplitude was more significantly lower at around −330 μV and declined to approximately 50% (approximately 160 μV) over the course of a year (**Fig 4B**). By the age of 3 months, scotopic b-wave response in the mutants was also significantly lower and declined to approximately 50% over a year (**Fig 4C**). Similarly, in photopic conditions significant changes in a- and b-wave amplitudes compared to controls were observed from 3 months of age onwards and declined by approximately 50% over the course of a year (**Fig 4D and 4E**).

These findings were consistent with the morphology of the retinal layers as observed via optical coherence tomography (OCT) (**Figs 4F–4H, S4A and S4B**) [38]. Highly significant changes in the thickness of the retina in *Ift20$^{null}$;Tyrp2-Cre* mice, compared to controls, were only observed from 6 months of age. This was predominantly driven by the loss of photoreceptor cells, noticeable by a significant reduction of the outer nuclear layer, consistently declining from the age of 5 months onwards (**Fig 4H**, black significance levels). This was accompanied by thinning of the POS layer, while inner segment thickness did not significantly decline until 10 months of age. Collectively, these measurements show the subsequent reduction of outer retinal layers over time. Despite these changes, visual acuity did not seem to be affected as measured via the OMR. Visual acuity thresholds remained stable at around 0.38 cyc/° in *Ift20$^{null}$; Tyrp2-Cre*, as well as in control *Ift20$^{+/+}$;Tyrp2-Cre* (**S4C Fig**).

The increasing pathogenicity of the RPE in *Ift20$^{null}$;Tyrp2-Cre* mice over time led us to examine macrophage/microglia in the subretinal space in older mice (1 to 2 years). We fluorescently labeled RPE flatmounts with the macrophage/microglia markers IBA1 and F4/80 (**S5B Fig**). We imaged RPE flatmounts and quantified the number of macrophage/microglia per mm$^2$ (**S5A Fig**). Combining all ages revealed significantly more macrophage/microglia in *Ift20$^{null}$;Tyrp2-Cre* animals (**S5B Fig**). These findings were confirmed in a separate experiment with a different cohort of mice (**S5C–S5E Fig**). Increased expression of these markers is a sign of elevated inflammatory and degenerative processes as a result of, and possibly contributing to, pathogenicity.

Combined, these data suggest a normally developing retina with relatively stable morphology and function in the first 2 months of age, after which we observe a progressive retinal degeneration and a decline in ERG responses.

In summary, all these observations strongly suggest that the ablation of primary cilia exclusively in the RPE results in a progressive ocular pathogenicity that leads to visual impairment. Loss of primary cilia only in the RPE causes defects in RPE maturation and RPE function, both of which ultimately affect photoreceptor health and cause their degeneration. Along with the progressive demise of retinal photoreceptors, we could show that ERG responses in *Ift20$^{null}$;Tyrp2-Cre* mice declined, leading to visual impairments of those mice.

## Discussion

Primary cilia dysfunction leads to a wide range of pathological phenotypes, collectively termed ciliopathies, with retinal degeneration being one of the most common. Ciliary mutations do

not only cause syndromic retinal disorders, but they also underlie numerous non-syndromic retinal dystrophies (https://sph.uth.edu/Retnet/). So far, most research on retinal ciliopathies has focused on the highly specialized primary cilium of the photoreceptors, which undoubtedly contribute significantly to disease progression. However, the contribution of dysfunctional primary cilia in other ocular cell types has not yet been comprehensively studied [2,6,7].

We recently showed that loss of *Bbs8*, a gene encoding for a component of the BBSome, which is required for ciliary trafficking, results in defects in RPE homeostasis and function [8]. Although it could be shown that defective cilia in the RPE affect RPE function, it must be considered that interaction with defective photoreceptors might be a leading driver to this effect. Thus, distinguishing whether defects in the RPE contribute to retinal defects is difficult. To be able to delineate the mutual influence of these 2 tissues, we aimed at dissecting the role of ciliary defects in the RPE by ablating primary cilia exclusively in the RPE, for which we used an RPE-specific *Ift20* conditional knockout mouse model [14,15].

Here, we demonstrate that ablation of cilia due to loss of IFT20 in the RPE leads to defects in RPE homeostasis and functionality, ultimately leading to retinal degeneration along with visual impairment. Although most cilia are retracted upon RPE maturation in rodents, embryonic ciliation is required for development and maturation of the RPE, which we have now shown has long lasting consequences. Importantly, adjacent retina was initially not affected since the morphology of the retina in TEM sections looked comparable to control, and ERG function was not significantly altered until 2 months of age. This suggests that our Cre driver is not leaky and that any subsequent loss in retinal function is a secondary consequence of disruption in the RPE. We conclude that the RPE phenotype likely precedes photoreceptor degeneration since the RPE displays a significant phenotype (already evident at P16 (Fig 3F and 3G)). In the context of ciliopathies, it must be noted that the loss of cilia function in the RPE is presumably less impactful than loss of ciliary function within the photoreceptor itself. However, this could be perceived as an added source of strain, further intensifying the stress on ocular tissues.

The choice of Cre driver is an important factor to consider. Many RPE-specific Cre drivers are under the control of late RPE genes, which only become active once the RPE has matured. However, since the cilium is expressed as early as E14.5 [7,19], we chose a Cre driver with early RPE Cre activity starting at E11.5 [15]. The majority of RPE cells retract their cilium post birth, with only 5% to 10% remaining ciliated in adult [19]. Since our Cre driver is already active during embryonic development, we are not able to address the question of what these remaining ciliated RPE cells are doing and how they might be contributing to RPE function post-development. For this we would need to use an inducible Cre driver to ablate ciliation only in the RPE at later time points, such as the *Tyrosinase-CreEr^T2* mouse line that we recently characterized [39].

We used the *Ift20^flox* mouse line to ensure that all aspects of ciliation would be ablated. In addition to IFT, which is essential for trafficking along the ciliary axoneme, IFT20 is also required for trafficking from the Golgi to the base of the cilium [17], thereby influencing very early stages of ciliogenesis. In contrast, floxed mouse lines with many other *IFT* genes still retain some residual ciliary machinery. It is also important to consider that many ciliary proteins have also been shown to exert non-ciliary functions, disruptions of which likely contribute to a cellular phenotype [40–43]. In particular, IFT proteins are known to interact with non-ciliary microtubules and motors and can affect processes of cell division [44]. These non-ciliary functions of IFT20 might also be playing a role in our model. The alternative function of IFT20 involving collagen secretion [45, 46] would be interesting to check in the context for the RPE, since collagen has been shown to be important in RPE function [47]. Similarly, it would be important to investigate the role of IFT20 in the Hippo Pathway in the context of the

RPE [48], since this pathway is also known to play a part in RPE cell fate determination [49,50].

While we cannot discount contributions from other non-ciliary phenotypes of IFT20 loss, it is highly likely that the phenotype in the RPE is largely coming from the loss of cilia. In general, all IFT20 phenotype models correlate strongly with other cilia mutants [18,51–53]. In particular with IFT74 and IFT88, which are also components of the IFTB trafficking complex [54–56]. Numerous differentially regulated transcripts that overlap with the differentially regulated BPs observed in the Bbs8 cilia mutant knockout RPE, particularly at P11 [57], were also detected by our RNA sequencing analysis. From an evolutionary angle, IFT20 is not found in any organisms without cilia, suggesting that its function is primarily related to the cilium [58]. In particular since IFT20 is a component of the IFTB complex, the most ancestral ciliary trafficking complex [59]. Finally, despite many studies showing IFTB proteins involved in cell division [44], our phenotype was not compatible with defects in cell division and spindle orientation. These defects would have been evident in the morphological patterning. However, overall, it is plausible that we are witnessing a compounded effect involving both developmental defects, leading to immature/pathogenic RPE, and subsequent effects resulting from the loss of IFT20 in adulthood. This effect may arise from the remaining few ciliated cells or from non-ciliary functions of IFT20. Careful examination and comparison of different RPE-specific cilia mutants at specific time points is needed to tease out the role of the cilium versus ciliary proteins in other cellular functions.

Of particular interest is the role of IFT20 and cilia in relation to the actin cytoskeleton. Ciliary signaling and ciliary proteins regulate various aspects of actin dynamics [60]. The RPE cannot be viewed alone, but as a functional unit closely associated with the POS. This interaction is mediated via the actin-based apical processes, modified microvilli, extending from the RPE apical surface. In our data, we have seen numerous indications that this close connection is disrupted. We observed changes in the retinal adhesion, decreased rates of phagocytosis, as well as structural changes ("gaps" at 1 month of age and excess material at 3 months of age) in the subretinal space. Subretinal "gaps" may be a result of the technical preparation of the tissue. Fixation artifacts can be increased in models of retinal degeneration due to loss of outer segment material and/or photoreceptor cells. Since we also showed reduced attachment between RPE and POSs, these abnormalities might be increased upon tissue preparation in the mutants. Nonetheless, they highlight a difference in tissue integrity.

The accumulated debris seen at 3 months of age might be excess microvilli-like structures or possibly non-phagocytosed POS, both of which could also be a consequence of disrupted interaction between the RPE and retina. In our phagocytosis assay, we were unfortunately not able to distinguish between bound and internalized POS. POS binding relies on short microvilli at the RPE apical surface encompassing aged POS tips exposing phosphatidylserines, indicating which parts need to be eliminated [61]. Retinal adhesion on the other hand involves long RPE microvilli and the whole length of POSs [62,63]. Hence, both functions being independent, decreased retinal adhesion is not automatically associated with decreased POS binding and, thus, internalization, but it might be the case as suggested by our results in this model.

The question remains whether the dysfunctional apical processes are a consequence of ciliary signaling and secondary cytoskeletal rearrangements or whether ciliary proteins have a direct effect on these actin networks. The modest changes in RPE cellular morphology, as seen via REShAPE analysis, might also be a consequence of cytoskeletal aberrations. Further examination of cellular morphology at later stages would be of interest, however, the older the tissue becomes, the higher the likelihood of secondary consequences due to retinal degeneration.

We were encouraged to see that the stimulus-related response of the RPE reflected by the different dcERG components was significantly reduced in our mutant mice from as early as 1

month of age, which indicates reduced or malfunction of the RPE. Light causes a hyperpolarization in the photoreceptors, resulting in a reduction in subretinal $K^+$. This leads to a hyperpolarization in the RPE apical membrane and a decrease in intracellular $K^+$ as the RPE tries to compensate the loss of $K^+$ in subretinal space. This is followed by a delayed hyperpolarization of the basal membrane. The RPE response is consequently not directly induced by light but reflects the responses to photoreceptor light response [32]. A decrease in the c-wave response suggests changes in inward rectifying $K^+$ conductance. The positive c-wave, which peaks several seconds after the light onset, consists of 2 underlying potentials, both involving inward rectifying $K^+$ conductance [64–66]. It is firstly comprised of a negative potential caused by the Müller cells and secondly of a much larger positive potential caused by the apical membrane of the RPE when measured at the cornea surface. Furthermore, a significant change in FO represents defects in the delayed basolateral membrane hyperpolarization [67]. The most significant change was observed in the LP response, which represents the activities of Cl-channels depolarizing the basal membrane [35,68].

The transcriptomic data offers first insights into mechanisms underlying pathogenic changes. Despite the limitations of tissue preparation and contamination of adjacent retina, which could be eliminated by doing single-cell sequencing, these data still offer valuable insights into molecular mechanisms. Perhaps unsurprisingly, many of the differentially expressed transcripts examined at the earliest time point (P11) were related to development, differentiation, and maturation. As the tissue ages, we found more changes in transcripts related to RPE function, which reflect an impaired apical membrane structure, possibly initiating EMT. There is mounting evidence that ablation of primary cilia triggers EMT [69,70]. Moreover, the association between loss of ciliation in the RPE and EMT is in line with our previous findings in congenital *Bbs8* knockout mice, in which we showed that loss of BBS8 induces EMT-like traits in the RPE, and is consistent with higher AR in mutant RPE cells as we see here [8]. The increased appearance of macrophages and microglia in the subretinal space in much older mutant animals provides additional support for the hypothesis implicating EMT-like processes. Our data, suggesting that lack of cilia might underlie a possible EMT phenotype, which precedes and is possibly the cause of photoreceptor degeneration, is in line with other recent findings that EMT in the RPE underlies retinal degeneration [71–73]. This might offer a valuable entry point into therapeutic interventions. Since these initial findings had not been done in a conditional RPE-specific knockout, it had been difficult to attribute the phenotype to ciliation in the RPE alone, yet our current findings support a direct link between these two. Additional mouse models using alternative floxed drivers for various cilia-related genes, as well as different RPE-specific Cre drivers that ablate ciliary proteins at various time points in RPE development and maturation would be helpful to pinpoint more precise roles of cilia and cilia proteins in all aspects of RPE biology.

In summary, we could demonstrate that loss of IFT20 ablates cilia in the RPE and leads to a pathogenic phenotype that has a significant impact on retinal degeneration leading to visual impairment. These data should be considered when formulating treatment strategies for retinal degeneration of ciliopathy patients.

## Materials and methods

### Animals

Mice were housed in a 12-h light/dark cycle. The morning after mating was considered as embryonic day (E) 0.5 and up to 24 h after birth was considered as postnatal day (P) 0. Animals were sacrificed by cervical dislocation. All animals were scarified at the same time of day, namely 3 h after light onset.

Mice with RPE-specific depletion of *Ift20* (*Ift20*[null]*;Tyrp2-Cre*) were generated by crossing *Ift20*[flox/flox] mice with *tyrosinase-related protein-2 (Tyrp2)-Cre* mice [14,15]. *Ift20*[flox/flox] mice possess LoxP sites in introns 1 and 3. Crossing these mice with *Tyrp2-Cre* mice leads to deletion of exons 2 and 3 and creates the *Ift20*[null] allele [14]. Homozygous *Ift20*[null]*;Tyrp2-Cre* mice were viable after birth. Genotyping for the *Ift20* alleles was done by PCR using following primers [14]: (A) 5′-ACTCAGTATGCAGCC-CAGGT-3′, (B) 5′-GCTAGATGCTGGGCGTA AAG-3′. To detect the *Cre* allele, following primers were used: (Cre-F) 5′-CAATGCTGTTTC ACTGGTTATG-3′, (Cre-R) 5′-CATTGCCCCTGTTTC-ACTACT-3′.

*B6.Cg-Gt(ROSA)26Sor*[tm14(CAG-tdTomato)Hze]*/J* reporter mice (JAX stock #007914; ([20]) referred to as *Ai14*) were crossed with *tyrosinase-related protein-2 (Tyrp2)-Cre* mice to visualize *Cre* expression. *Ai14* mice possess a floxed stop cassette upstream of the CAG promoter-driven red fluorescent protein tdTomato. Following Cre-mediated recombination the stop cassette is excised, allowing expression of tdTomato.

## Ethics statement

All animal experiments had ethical approval from the Landesuntersuchungsamt Rheinland-Pfalz under the approval number (23177-07/G21-1-005) and were performed in accordance with the institutional guidance for care and use of laboratory animals. Animal maintenance and handling was performed in line with Federation for Laboratory Animal Science Associations (FELASA) recommendations.

## Antibodies

For immunofluorescence, the following primary antibodies were used: anti-ARL13B (rb, 1:800, Proteintech, #17711-1-AP), anti-GM130 (mMAb, 1:100, BD Biosciences, #610822), anti-GT335 (mMAb, 1:800, Adipogen, #AG-20B-0020), anti-IFT20 ([74]), anti-Iba1(rb, 1:100, FUJIFILM Wako Pure Chemical Corporation, #019–19741) or anti-Iba1(1:300, Abcam, #ab153696), anti-F4/80 [CI:A3-1] (mMAb, 1:100 Abcam, #ab6640) or anti-F4/80 (1:150, AbD Serotec, # MCA497GA). These antibodies were detected using the appropriate AlexaFluor (AF) -488-, and -555-conjugated (1:400; Molecular Probes) secondary antibodies. Anti-Zonula Occludens-1 (ZO-1) was directly conjugated with AF-488 (1:100, ZO-1-1A12, Invitrogen, 339188) and Phalloidin was directly conjugated with AF-647 (1:40, Cell Signaling Technology, #8940).

## RPE flatmount preparation and immunohistochemistry

Mice were sacrificed and the eyes were enucleated. Anterior segment, lens and retina were removed, and eyecups were fixed with 4% paraformaldehyde (PFA) in 1× phosphate-buffered saline (PBS) for 1 h, followed by 3 washing steps with 1× PBS. To reduce PFA-induced autofluorescence, eyecups were incubated with 50 mM $NH_4Cl$ for 10 min, before permeabilizing with 1× PBS with 0.3% Triton-X (TX) (PBS-TX) for 1 h and blocking with blocking buffer (0.1% ovalbumin, 0.5% fish gelatin in 1× PBS-TX) for 1 h. Following this, eyecups were incubated with primary antibodies overnight at 4°C. Eyecups were washed 3 times with 1× PBS-TX, followed by incubation with secondary antibodies, directly conjugated antibodies, and DAPI (Carl Roth) for 2 h in dark conditions. Post staining, 2 washing steps with PBS-TX and 1 with 1× PBS for 20 min each were performed, and eyecups were mounted with Fluoromount-G (SouthernBiotech) and examined using a Leica DM6000 B microscope. Deconvolution (BlindDeblur Algorithm, 1 iteration step) and maximum projection were performed using Leica imaging software (Leica, Bensheim, Germany). Images were processed via Fiji using color correction and contrast adjustment ([75]).

For Microglia/Macrophage analysis, RPE flatmounts were co-stained for IBA1 and F4/80 and scanned using the Zeiss Axioscan. IBA1 and F4/80 positive cells were manually counted across the whole flatmount using ImageJ. Values are represented as mean of total cells counted by 3 individuals.

## Cell morphology and pigmentation assessment

RPE flatmounts were stained for F-actin or ZO-1 and scanned using the Zeiss Axioscan. Cells were segmented using REShAPE analysis, a machine learning based software previously published by Ortolan and colleagues [26]. Data for cell size, AR, and hexagonality is displayed as Median ± SD, number of neighbors as Mean ± SD.

To quantify pigmentation, flatmount preparations from P0 mice were imaged via differential interference contrast (DIC) overlayed with immunohistochemistry for cell nuclei (DAPI) and cell borders (ZO-1). Images were converted to 8-bit grayscale. A mask was calculated with all regions where pixel values are below a threshold (10). The total size of the image was calculated in $\mu m^2$ and the masked area per $\mu m^2$ was computed by dividing the total area below threshold by the image size in $\mu m^2$.

## Retinal adhesion assay

Retinal adhesion assay was performed as described in by Schneider and colleagues [8], previously described by Nandrot and colleagues [37]. Mice were sacrificed, eyes were enucleated and transferred into 1× Hank's balanced salt solution containing $Ca^{2+}$ and $Mg^{2+}$ (HBSS+) (Gibco, #14025–092). Adjacent tissues, cornea, and lens were removed. One eyecup at a time was transferred into a dry, empty dish and cut radially around the cornea. The neural retina was then peeled off using forceps. Retinae were lysed individually in 50 mM Tris (pH 7.5), 2 mM EDTA, 150 mM NaCl, 1% Triton X-100, 0.1% SDS, and 1% NP-40, freshly supplemented with 1% protease and phosphatase inhibitors via sonication for 10 s on ice. After centrifugation (5 min, 21130 rcf, 4˚C), lysates and pellets were kept separate on ice and protein content of the lysates was quantified using bicinchoninic acid (BCA) assay.

For melanin dissolution, the pellets were washed in 100 μl 50% ethanol and 50% diethylether (10 min, 21130 rcf). Supernatant was discarded, pellets were dissolved in 150 μl 20% DMSO, 2 M NaOH and incubated for 30 min at 60˚C. To quantify melanin concentration, absorbance of samples and commercially available melanin (Sigma-Aldrich, #M0418) dissolved in 20% DMSO, 2 M NaOH at defined concentrations were measured at 490 nm. Individual melanin concentrations were normalized to the corresponding protein concentration to calculate the concentration of melanin per milligram of protein.

## Transmission electron microscopy (TEM)

Mouse eyes were fixed in 0.1 M cacodylate-buffered fixative containing 2.5% glutaraldehyde and 0.1 M sucrose for 1 h at room temperature. After 10 min of fixation, the anterior segment was removed. The eyecups were then postfixed in 2% osmium tetroxide (in 0.1 M cacodylate buffer) for 1 h at room temperature followed by dehydrating steps using a graded ethanol series (30% to 100%). After embedding in epoxy resin, eyes were cut and processed for TEM using standard EM procedures and analyzed using a FEI Tecnai 12 BioTwin transmission electron microscope.

## Scanning electron microscopy (SEM) of embryonic eyecups

Pregnant dams were sacrificed, and the gravid uteri were removed and placed into 1× PBS (room temperature). To expose the embryos, the uterine wall and fetal membranes were

dissected carefully in order not to harm the embryos. Following this, the embryos were sacrificed, and eye dissection was performed in 1× HBSS, supplemented with 0.1 M HEPES (HBSS-H). Anterior segment, lens, and retina were removed, and eyecups were fixed with HBSS-H containing 4% PFA, 2.5% glutaraldehyde, and 10 mM $CaCl_2$ overnight at 4°C. Following this, 3 washing steps with 1× PBS supplemented with 0.1 M HEPES (1× PBS-H) were performed. The eyecups were then postfixed in 1% osmium tetroxide (in 1× PBS-H) for 1 h at room temperature, followed by 3 washing steps of 5 min each using distilled water. Next, dehydrating steps using a graded ethanol series (30% to 100%) were performed. Following dehydration, the eyecups were dried using a critical point dryer (Bal-Tec CPD 030). Dried eyecups were mounted carefully onto conductive carbon adhesive on top of SEM studs under a stereo microscope using a fine brush and forceps and sputter coated with gold (115 s). Samples were stored in a desiccator until microscopy.

## RPE cell isolation

RPE cell isolation was performed as described by Schneider and colleagues [8]. RPE pellets were dissolved in 100 μl TRIzol Reagent (Invitrogen), snap-frozen in liquid nitrogen and kept at −80°C.

## RNA isolation

RPE cells were homogenized in TRIzol Reagent (Invitrogen) using a pestle and TRIzol Reagent was added to a final volume of 500 μl. For RNA extraction, TRIzol Reagent was used according to manufacturer's recommendations and RNA was stored at −80°C until usage.

## QuantSeq 3′ mRNA sequencing library preparation

Preparation of libraries, 3′ mRNA sequencing data processing and analysis were performed as described by Schneider and colleagues [8].

## Data visualization/graphical image

Heatmaps (S3B–S3D Fig) were generated using custom annotated scripts. Schematic of experimental procedure of retinal adhesion assay (Fig 3F) was generated by hand using Inkscape version 1.0.2 (e86c8708, 2021-01-15).

## qPCR validation of QuantSeq 3′ mRNA sequencing

qPCR was performed as described by Schneider and colleagues [8] using iTaq Universal SYBR Green Supermix (Bio Rad, #1725121) on the QuantStudio 3 Real-Time PCR System.

Primers: Housekeeping Tbp_Fwd: `TGTATCTACCGTGAATCTTGGC`, Tbp_Rev: `CCAGAAC TGAAAATCAACGCAG`, Mfge8 F: `AGACATGGAACCTGCGTG`, Mfge8 R: `ATTCCTGTCACTT GCCTCTG`.

## Primary mouse RPE cell culture

RPE cells from 9- to 12-day-old animals were sequentially isolated as previously described and adapted as described below ([76]). Briefly, eyecups without their lens were digested for 45 min at 37°C with 1.5 mg/ml hyaluronidase (Sigma), and neural retina was gently peeled from the eyecup. RPE sheets were peeled from the Bruch's membrane after a 45 min digestion step at 37°C with 1 mg/ml trypsin (Invitrogen) and seeded into the wells of a 384-well plate after a last trypsin digestion step. Cells were grown to confluence for 5 to 10 days in MEM α modification (Sigma), supplemented with 5% fetal bovine serum (FBS, HyClone, GE Healthcare Life

Sciences), 1% N1 medium supplement (Sigma), 0.5 mM glutamine, 1% non-essential amino acids (Gibco), 1% penicillin/streptomycin (Gibco), 250 mg/L taurine, 20 μg/L hydrocortisone, and 0.0130 μg/L triiodo-thyronin at 37˚C, 5% $CO_2$ ([77]).

### In vitro phagocytosis assay

POS were isolated from porcine eyes, obtained fresh from the slaughterhouse, and covalently labeled with fluorescein isothiocyanate (FITC) dye (Invitrogen) for in vitro phagocytosis assays as previously described [38]. Cultured primary mouse RPE cells were challenged with approximately 10 FITC-POS per cell for 1.5 h. Great care was taken to load the same amount of POS in each well. For this $1.1 \times 10E7$ POS were resuspended into 300 μl medium and 14 μl (i.e., $5.13 \times 10E5$ POS/well) were applied in each well of a 384-well plate containing either control or mutant RPE primary cells. Nonspecifically tethered POS were removed with 3 thorough washes in 1× PBS containing 1 mM $MgCl_2$ and 0.2 mM $CaCl_2$ (PBS-CM). Cells were fixed with 4% PFA, PFA was quenched with 50 mM $NH_4Cl$, and nonspecific sites were blocked using 1% BSA in PBS-CM. Tight junctions were labeled using an anti-ZO-1 antibody (rb, Invitrogen), followed by an incubation with a mixture of the appropriate AF-594 secondary antibody and Phalloidin, directly conjugated with FluoProbes 647H (Interchim, #FP-BZ9630), to label the actin cytoskeleton. Nuclei were counter-stained with Hoechst 33258 (Invitrogen). FITC-POS and Hoechst-labeled nuclei were quantified by fluorescence plate reading using the Morphology v9 plug-in to identify nuclei and delimitate cell boundaries (HCS Studio Cell Analysis Software, Cellomics Arrayscan VTI HCS reader, Thermo Scientific). Quantification: Cell junction markers were used to delineate cell borders. When the labeling was not clear enough cells were not counted. POS were counted per identified cells and ratios of numbers of POS/cell were calculated. Upon immunofluorescent assays in primary cultures, there are instances where nuclei also exhibit weak fluorescence in the green channel. Our quantification software effectively eliminates faint nuclei-associated green signals by applying specific size and brightness criteria. Quantification was performed on 3 independent assays of 4 to 9 wells each, and significance was assessed with the Welch-corrected parametric unpaired Student $t$ test ($P < 0.05$ *).

### Electroretinography (ERG)

Full-field ERGs were recorded in mice at the age of 1 to 12 months. Mice were dark adapted for >12 h. Mice were anesthetized by intraperitoneal (i.p.) injection of a ketamine (87.5 mg/kg)/xylazine (12.5 mg/kg) cocktail and pupil dilator was applied (1% tropicamide/2.5% phenylephrine). ERG responses of the outer retina were recorded on a full-field stimulator (Espion E3 ColorDome; Diagnosys, LLC, Lowell, Massachusetts, United States of America) with a Gold electrode attached to the corneal surface of both eyes referenced to needle electrodes (Diagnosys) between the ears and in the tail. For electrical contact and corneal integrity, a drop of 2% methylcellulose was applied to each eye. Mice were placed on a heating pad (37˚C) under the ColorDome (Diagnosys) and subjected to 6 steps of strobe flashes of increasing stimulus intensities (dark adapted: 0.0001, 0.001, 0.01, 0.1, 1, and 10 cd*s/m²) followed by 2 min of light adaptation and flashes of 0.3, 1, 3, 10, 30, and 100 cd*s/m², concluded by a 10-Hz flicker stimulus of 100 cd*s/m². The a-wave was determined by measuring the peak of the first negative wave. The b-wave was calculated from the trough of the a-wave to the peak of the first positive wave or from baseline if no a-wave was present.

### Optical coherence tomography (OCT)

OCT (Bioptigen, Research Triangle Park, North Carolina, USA) was performed subsequent to full-field ERG measurement in order to reduce burdens caused by anesthesia. The OCT image

was captured using a rectangular volume scan (20 B-scan 1 frame). Images were imported as stacks in ImageJ (National Institutes of Health, Bethesda, Maryland, USA). The StackReg plugin was used to remove image distortion due to respiration of the animal ([78]). Slices were then merged using "Z project" and "Sum Slices." Retinal layers were measured manually 500 μm from the center of the optic nerve ([79]). Data processing was done using Matlab. Statistical comparisons of $Ift20^{null};Tyrp2$-$Cre$ versus $Ift20^{+/+};Tyrp2$-$Cre$ mice were done using the Holm–Sidak test ([80,81]) and statistical comparison of P25 versus other ages was performed using the Dunnet's multiple comparison ([82]), both followed with a one-way ANOVA (significance levels: >0.05 not significant (ns), <0.05 *, <0.01 **, <0.001 ***).

## Optomotor response (OMR)

OMR was recorded using the quantitative OMR setup (qOMR, Phenosys, Berlin, Germany), which allows for quantitative measurement and analysis of the behavior of freely moving mice. Vertical stripes of 13 different spatial frequencies between 0.0125 and 0.5 cyc/deg were presented on a rotating virtual sphere surrounding the animal. To keep the spatial frequency constant, the mouse was video tracked to automatically maintain the distance between the animal and the virtual sphere. Therefore, the perceived spatial frequency was maintained ([83,84]). Each stimulus was presented for 60 s. The automated tracking was used to quantitatively evaluate all experiments. Stimulus-correlated head movements were determined and the ratio of movements within a velocity range of 2 to 14 deg/s in the correct direction, divided by movements in the same range in the incorrect direction, were calculated and defined as the OMR (OMRindex). Each set of stimuli was presented 3 times in a pseudo-randomized order with resting time of a minimum of 1 h between trials. Data analysis was done using the quantitative OMR software and Matlab. To determine the range of stimulus uncorrelated activity of all animals (baseline), we calculated the interquartile interval for the OMR for all mice at 0.5 cyc/deg, a spatial frequency not perceivable by the animals (OMRindex, 1.15; **S4 Fig,** yellow areas) ([85]). Thus, the spatial frequency threshold was calculated as the intersection of polynomial fit (third degree) with this baseline interval.

## Direct-coupled electroretinography (DC-ERG)

DC-ERGs were recorded from anesthetized (same conditions as in "Electroretinography") P25 mice using microelectrode holder half-cells (holder/pellet) (MEH34515, WPI) with borosilicate glass capillary tubes OD: 1.5 mm ID: 0.84 mm (1B150F-3, WPI) filled with Hank's buffered salt solution (15266355, Gibco). Electrodes remained in direct contact with the cornea through 2% methylcellulose. Electric contact was monitored for 10 min prior to the recording to facilitate steady-state conditions. Data was analyzed, smoothed using a moving average filter spanning 100 ms and displayed using Matlab (The MathWorks Inc., Natick, Massachusetts, USA). The linear drift was corrected by subtracting the best fit line through the 300 ms preceding light onset. DC-ERG components were then measured as follows: c-wave: maximum in the 500 ms after light onset, FO: peak of c-wave to minimum in 1,500 ms after stimulus onset, LP: difference between FO and response at stimulus offset (4,500 ms), Off: minimum after stimulus offset. Statistical analysis was done using the unpaired two-tailed $t$ test (significance levels: >0.05 not significant (ns), <0.05 *, <0.01 **, <0.001 ***).

## Supporting information

**S1 Fig. Characterization of mouse lines.** (**a**) Representative fluorescent images of E16.5 RPE flatmounts stained for IFT20 (red) and *cis*-Golgi matrix protein GM130 (green). Staining for F-Actin (magenta) was used to visualize the cytoskeleton, whereas DAPI was used to stain

nuclear DNA. E16.5 *Ift20^{null};Tyrp2-Cre* RPE showed less IFT20 staining compared to controls. Scale bars: 10 μm. (**b**) Quantification of IFT20 positive cells in E16.5 RPE revealed that *Ift20-^{null};Tyrp2-Cre* RPE showed near to no IFT20 staining (5.3% $n = 4$ (1,222 cells)) compared to control (98.5% $n = 4$ (1,236 cells)), confirming the knockout. Statistical analysis was performed using ROUT test (Q = 0.1%) before using unpaired $t$ test ($p < 0.001$). Median: *Ift20^{+/+};Tyrp2-Cre* 98.5%, *Ift20^{null};Tyrp2-Cre* 5.3%. (**c**) Representative retina cross section of 1-month-old *Ift20^{null};Tyrp2-Cre* mouse crossed with a tdTomato reporter mouse. F-Actin staining visualized by Phalloidin staining (light blue) and DAPI for nuclei (blue). tdTomato expression (red) representing Cre activity was detectable only in RPE. Scale bar: 20 μm. Significance levels: >0.05 not significant (ns), <0.05*, <0.01**, <0.001***. Box plot: Box limits represent the first and third quartile, the central line shows the median and the whiskers indicate the 5th and 95th percentile. Numerical data can be found in S7 Table.
(TIFF)

**S2 Fig. Deletion of *Ift20* in the RPE results in defective RPE morphology.** (**a**) HE-stained cross section through the mouse eye reveals subretinal gaps in the outer layers of *Ift20^{null}; Tyrp2-Cre* retina compared to *Ift20^{+/+};Tyrp2-Cre*. (**b**) Quantification of subretinal gaps revealed a median of 0.5 gaps in *Ift20^{+/+};Tyrp2-Cre* mice and 9 gaps in *Ift20^{null};Tyrp2-Cre* mice per semithin section quantified (*Ift20^{+/+};Tyrp2-Cre* $n = 9$ sections, 4 eyes, 3 mice, *Ift20^{null}; Tyrp2-Cre* $n = 9$ sections, 4 eyes, 3 mice). (**c**) Representative SEM images of P0 RPE flatmounts. No differences in microvilli morphology were observed between P0 *Ift20^{null};Tyrp2-Cre* and *Ift20^{+/+}Tyrp2-Cre* RPE. Scale bar: 5 μm. (**d**) Representative TEM images of eye sections from 1-month-old mice showing a discontinuous Bruch's membrane (red brackets) at *Ift20^{null}; Tyrp2-Cre* RPE cells. Scale bar: 2 μm. (**e**) Representative TEM images of *Ift20^{null};Tyrp2-Cre* show abnormal pigmentation at 1 month and 4 months. Two pigmented RPE cells flank a cell almost completely devoid of melanosomes at 1 month. At 4 months of age, we observed an RPE cell that appeared devoid of melanosomes in the cell body, which was flanked by an RPE cell with an excessive accumulation of melanosomes. Scale bar = 10 μm. (**f**) Flatmount preparations from P0 mice imaged via differential interference contrast (DIC) overlayed with immunohistochemistry for cell nulcei (DAPI) and cell borders (ZO-1). Images were converted to 8-bit grayscale and masked to calculate pigmentation area. (**g**) Quantification of pigmentation in terms of area covered. No significant differences could be detected between control and mutant animals. Median: *Ift20^{+/+};Tyrp2-Cre* 7.5 $\mu^2$ area of intensity below threshold, *Ift20^{null}; Tyrp2-Cre* 7.9 $\mu^2$ area of intensity below threshold. *Ift20^{+/+};Tyrp2-Cre* ($n = 9$ eyes (44 images)), *Ift20^{null};Tyrp2-Cre* ($n = 6$ eyes (27 images)). Significance levels: >0.05 not significant (ns), <0.05*, <0.01**, <0.001***. Box plots: Box limits represent the first and third quartile, the central line shows the median and the whiskers indicate the 5th and 95th percentile. Numerical data can be found in S7 Table.
(TIFF)

**S3 Fig. Deletion of *Ift20* in the RPE leads to changes in RPE-specific gene expression.** (**a**) Top: Identification of the biological processes underlying the effect of the deletion of *Ift20* via gene ontology analysis. The top 10 down-regulated significant biological processes (BPs) in *Ift20^{null};Tyrp2-Cre* vs. *Ift20^{+/+};Tyrp2-Cre* are shown. The Enrichment score for each BP cluster is plotted on the y-axis. Bottom: Identification of the biological processes underlying the effect of the deletion of *Ift20* via gene ontology analysis. The top 10 up-regulated significant biological processes (BPs) in *Ift20^{null};Tyrp2-Cre* vs. *Ift20^{+/+};Tyrp2-Cre* are shown. The Enrichment score for each BP cluster is plotted on the y-axis. (**b–d**) Heatmaps showing DEGs associated with transmembrane transport (**b**), cytoskeleton organization (**c**), and tissue remodeling (**d**).

Data has been deposited in the GEO database (GSE144724).
(TIFF)

**S4 Fig. Retinal thickness of control mice unchanged over time and loss of IFT20 in the RPE does not affect the optomotoric reflex.** (**a**) Representative images of in vivo OCT scans of *Ift20^{+/+};Tyrp2-Cre* mice from 1 to 12 months. No overall change in could be seen over time. Scale bar: 25 μm. (**b**) Representative Images of in vivo OCT scans of *Ift20^{null};Tyrp2-Cre* mice compared to *Ift20^{+/+};Tyrp2-Cre* at monthly intervals from 1 to 12 months. Scale bar: 25 μm. (**c**) Optomotor response curves of *Ift20^{+/+};Tyrp2-Cre* and *Ift20^{null};Tyrp2-Cre* mice. Single measurements (purple dots) show a normal flicker. Visual acuity thresholds remained stable at around 0.38 cyc/° (dashed line, spatial frequency threshold) in both, *Ift20^{+/+};Tyrp2-Cre* and *Ift20^{null};Tyrp2-Cre* mice over the time of 12 months. For all experiments $n > 4$ animals. Numerical data can be found in S7 Table.
(TIFF)

**S5 Fig. Macrophages in RPE flatmount.** (**a**) Representative images of RPE flatmounts of *Ift20^{+/+};Tyrp2-Cre* and *Ift20^{null};Tyrp2-Cre* at 22 and 24 months of age. Macrophages were stained by markers IBA1 (green) and F4/80 (red). (**b**) Quantification of macrophage numbers per mm$^2$ separated by age. Median: 12–18m *Ift20^{+/+};Tyrp2-Cre* 1 macrophage/mm$^2$, *Ift20^{null};Tyrp2-Cre* 4.18 macrophages/mm$^2$. 22–24m *Ift20^{+/+};Tyrp2-Cre* 2.5 macrophages/mm$^2$, *Ift20^{null};Tyrp2-Cre* 4.31 macrophages/mm$^2$. Statistical analysis was performed using the unpaired two-tailed *t* test. 12–18m *Ift20^{+/+};Tyrp2-Cre* $n = 6$ eyes 7 images, *Ift20^{null};Tyrp2-Cre* $n = 4$ eyes 7 images. 22–24m *Ift20^{+/+};Tyrp2-Cre* $n = 4$ eyes 7 images, *Ift20^{null};Tyrp2-Cre* $n = 6$ eyes 7 images. (**c–e**) Combining all ages revealed a significant difference in macrophage number between *Ift20^{+/+};Tyrp2-Cre* and *Ift20^{null};Tyrp2-Cre*. (**c**) Representative images of entire RPE flatmount of *Ift20^{+/+};Tyrp2-Cre* and *Ift20^{null};Tyrp2-Cre* at 24 months of age. Macrophages were stained by markers IBA1 (green) and F4/80 (red). Median: *Ift20^{+/+};Tyrp2-Cre* 7.8 macrophages/mm$^2$, *Ift20^{null};Tyrp2-Cre* 20.9 macrophages/mm$^2$. Statistical analysis was performed using the unpaired two-tailed *t* test. *Ift20^{+/+};Tyrp2-Cre* $n = 9$ flat mounts, *Ift20^{null};Tyrp2-Cre* $n = 9$ flat mounts. Scale bar: 20 μm. Significance levels: $>0.05$ not significant (ns), $<0.05$ *, $<0.01$ **, $<0.001$ ***. Box plots: Box limits represent the first and third quartile, the central line shows the median and the whiskers indicate the 5th and 95th percentile. Numerical data can be found in S7 Table.
(TIFF)

**S1 Table. RNA-seq analysis of RPE cells isolated from *Ift20^{null};Tyrp2-Cre* and *Ift20^{+/+};Tyrp2-Cre* mice at P11, P29, and 3 months of age.** The differentially expressed genes (DEGs) highlighted in green represent up-regulated genes while the genes highlighted in red represents down-regulated genes.
(XLSX)

**S2 Table. Comparison of differentially expressed RPE signature genes between *Ift20^{null;-}Tyrp2-Cre* and *Ift20^{+/+};Tyrp2-Cre* mice at P11, P29, and 3 months of age.**
(XLSX)

**S3 Table. Enrichment analysis for biological processes (BP) using the differentially expressed genes either up- or down-regulated between *Ift20^{null};Tyrp2-Cre* and *Ift20^{+/+};Tyrp2-Cre* mice at P11, P29, and 3 months of age.** The threshold for statistical significance of GOEA was FDR$<0,1$ and Enrichment Score $\geq 1.5$.
(XLSX)

**S4 Table. List of 50 genes that were consistently up-regulated (green) and 16 genes were consistently down-regulated (red), between *Ift20^null;Tyrp2-Cre* and *Ift20^+/+;Tyrp2-Cre* mice across all 3 data points.**
(XLSX)

**S5 Table. Genes associated with transmembrane transport, cytoskeletal reorganization, and tissue remodeling shown in the heatmaps on S3B, S3C and S3D Fig.**
(XLSX)

**S6 Table. Numerical data for all data sets contained in Figs 1–4.**
(XLSX)

**S7 Table. Numerical data for all data sets contained in S1, S2, S4 and S5 Figs.**
(XLSX)

## Acknowledgments

This manuscript is dedicated to the memory of Elisabeth Sehn and her unwavering dedication to and fascination of electron microscopy and scientific exploration. The authors also wish to thank Petra Gottlöber for her technical assistance. Further, the authors would also like to thank A. Potey from the high-throughput screening platform of the Institut de la Vision (Paris) for her technical assistance in phagocytosis assays quantification. Finally, the authors are grateful to Diego Carrella (TIGEM Bioinformatics Core) for the Heatmap generation.

## Author Contributions

**Conceptualization:** Viola Kretschmer, Sandra Schneider, Peter Andreas Matthiessen, Kapil Bharti, Ivan Conte, Helen Louise May-Simera.

**Data curation:** Dominik Reichert, Nathan Hotaling, Rossella De Cegli.

**Formal analysis:** Nathan Hotaling, Rossella De Cegli.

**Funding acquisition:** Helen Louise May-Simera.

**Investigation:** Viola Kretschmer, Sandra Schneider, Peter Andreas Matthiessen, Dominik Reichert, Nathan Hotaling, Gunnar Glasßer, Ingo Lieberwirth, Rossella De Cegli, Ivan Conte, Emeline F. Nandrot, Helen Louise May-Simera.

**Methodology:** Viola Kretschmer, Sandra Schneider, Peter Andreas Matthiessen, Dominik Reichert, Ingo Lieberwirth, Kapil Bharti, Rossella De Cegli, Emeline F. Nandrot, Helen Louise May-Simera.

**Project administration:** Viola Kretschmer, Ivan Conte.

**Resources:** Kapil Bharti, Ivan Conte.

**Supervision:** Kapil Bharti, Helen Louise May-Simera.

**Visualization:** Viola Kretschmer, Sandra Schneider, Peter Andreas Matthiessen, Emeline F. Nandrot.

**Writing – original draft:** Viola Kretschmer, Sandra Schneider, Helen Louise May-Simera.

**Writing – review & editing:** Viola Kretschmer, Sandra Schneider, Peter Andreas Matthiessen, Rossella De Cegli, Ivan Conte, Helen Louise May-Simera.

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
