## [Editor Report · Decision Letter 0]

17 Dec 2022

Dear Dr May-Simera, 

Thank you for submitting your manuscript entitled "Ablation of primary cilia exclusively in the RPE leads to retinal degeneration and visual impairment." for consideration as a Research Article by PLOS Biology.

Your manuscript has now been evaluated by the PLOS Biology editorial staff as well as by an academic editor with relevant expertise and I am writing to let you know that we would like to send your submission out for external peer review.

Once your full submission is complete, your paper will undergo a series of checks in preparation for peer review. After your manuscript has passed the checks it will be sent out for review. To provide the metadata for your submission, please Login to Editorial Manager (https://www.editorialmanager.com/pbiology) within two working days, i.e. by Dec 20 2022 11:59PM.

Kind regards,

Ines

--

Ines Alvarez-Garcia, PhD

Senior Editor

PLOS Biology

---

## [Decision Letter · Decision Letter 1]

28 Feb 2023

Dear Dr May-Simera,

Thank you for your patience while your manuscript entitled "Ablation of primary cilia exclusively in the RPE leads to retinal degeneration and visual impairment." was peer-reviewed at PLOS Biology. Please accept my sincere apologies for the delay in sending you our decision. The manuscript has now been evaluated by the PLOS Biology editors, an Academic Editor with relevant expertise, and by four independent reviewers. 

The reviews are attached below. As you will see, the reviewers find the conclusions of the manuscript important, however they also raise several concerns that would need to be addressed in order for us to consider publication. They think that the impact that RPE cilia defects may play in human ciliopathies remains unclear and they all propose several experiments to strengthen these results. In addition, the reviewers ask for several clarifications, missing controls, quantification of some of the experiments and think that restructuring the text will significantly improve the flow of the manuscript.

After discussing the reviews with the Academic Editor, we would like to invite you to revise the work to thoroughly address the reviewers' reports. However, given the limitation in mechanistic insights, we would like to consider the manuscript as a Short Report. The main change that you would have to do to convert the manuscript into this format would be to reduce the number of main figures to four, making the rest supplementary figures (https://journals.plos.org/plosbiology/s/what-we-publish#loc-short-reports).

Given the extent of revision needed, we cannot make a decision about publication until we have seen the revised manuscript and your response to the reviewers' comments. Your revised manuscript is likely to be sent for further evaluation by all or a subset of the reviewers.

**IMPORTANT - SUBMITTING YOUR REVISION**

3. Resubmission Checklist

a) *PLOS Data Policy*

b) *Published Peer Review*

d) *Blurb*

Please also provide a blurb which (if accepted) will be included in our weekly and monthly Electronic Table of Contents, sent out to readers of PLOS Biology, and may be used to promote your article in social media. The blurb should be about 30-40 words long and is subject to editorial changes. It should, without exaggeration, entice people to read your manuscript. It should not be redundant with the title and should not contain acronyms or abbreviations. For examples, view our author guidelines: https://journals.plos.org/plosbiology/s/revising-your-manuscript#loc-blurb

Sincerely,

Ines

--

Ines Alvarez-Garcia, PhD

Senior Editor

PLOS Biology

Reviewers' comments

Rev. 1:

The manuscript by Kretschmer et al is an interesting study regarding the importance of IFT20 in RPE cells to retinal health. The RPE specific knockout of IFT20, a gene known to be crucial for the maintenance of primary cilia, results in a slow, progressive retinal degeneration associated with defects in RPE cells. The data presented are extensive and mostly of high quality. This study does a nice job in quantification of observed phenotypes. The major fault of the paper is that the phenotype induced by knockout of IFT20 occurs throughout the adult life of the mouse (very slow), when few, if any, RPE cells have a primary cilia. How could the effects of IFT20 knockout be exerted through ciliary defects if RPE do not have cilia to begin with? This point is essentially not addressed. Further, the potential non-ciliary functions of IFT20 are not mentioned or explored. This study should be revised to address these issues and present the data about the phenotype associated with IFT20 deletion from the RPE cells, rather than presume that the effects are through cilia ablation.

Concerns and comments:

1. Figure 1h, please indicate the age of the mouse analyzed either in the results section or in the figure legend.

2. Figure 2e, nowhere in the manuscript is it described how melanin concentration was obtained. Please describe in the results, methods and/or figure legend. Also, a whole retina measurement of melanin should be performed to consider the possibility that there is less melanin to begin with in the IFT20 KO RPE. The image from Fig. 1e suggests this may be the case. Later in the manuscript, the variation in melanosomes is demonstrated (Supplement Fig 2) which bolsters the idea that there could be a reduction in total melanin causing this effect. A formal quantification of total melanin should be performed.

3. Figure 2f, Is one of these the KO? They are both labeled IFT20+/+;Tyrp2-Cre. In these phagocytosis assays, how do you differentiate between POS that are phagocytosed with POS that are just adhered to the RPE? In Fig. 2e, you claim that the retina adheres more gently to the RPE which is consistent with you measuring a reduced binding of POS to the RPE in your phagocytosis assays.

4. Figure 4, when looking at the engulfment of POS by the RPE, what time of day were the mice sacrificed? Ideally, this should be noted in the methods/figure legend and the controls must have been sacrificed at the same time as the experimental mice.

5. Figure 4h, The term "microvilli depth" should not be used because it implies that you are measuring the length of microvilli. In fact, you have measured the space between the RPE apical surface and the tips of the photoreceptor outer segments. Perhaps it could be called "RPE to POS distance"

6. Line 220 - The title of this section says "single-cell analysis" which is a commonly used term for analysis of single-cell sequencing data. I suggest changing this title to avoid confusion.

7. Line 239 - 240 - The statement that RPE cell patterning defects could not be due to cell loss does not make sense. The cell area gets very slightly bigger. Does the retina and/or RPE layer get proportionally bigger as compared to WT? Probably not, but the authors could measure. If the RPE layer/retina stays the same size, then the number of RPE cells must be dropping as compared to WT. The larger individual RPE cells, and therefore lower RPE cell density, implies that some have died, leaving room for individual cell growth. Looking at the RPE cell size/density at a much later timepoint would help to answer this question. Because the reduction in RPE cell size is so small, the putative loss of RPE cells would also have to be small during the studied timeframe to P90. Given that caspase activation would happen on the order of minutes, it would likely be impossible to catch these rare events happening at any given time. Therefore, the authors cannot rule out that RPE cell death is contributing to cellular morphology changes. The caspase data should be removed from the paper.

8. The figure legend for Figure 6 is swapped with Figure 5.

9. Line 255 - The text should reference Figure 7b here.

10. Line 745 - Figure 8g is microscopy images, not western blot as described in the legend.

11. Line 346 - The correct point brought up in Discussion that ciliary proteins may have non-ciliary roles is essentially not addressed in this paper. The disease observed by the authors with IFT20 KO RPE is very slow and occurs over a timeframe when the vast majority of RPE do not have cilia. How could a ciliary defect cause this long-term pathology in cells that lost their cilia during development? Specific non-ciliary roles of IFT20 are not discussed, despite a seemingly abundance of literature on the topic (see PMID: 33072760 for recent review). The concept that IFT20 is doing something important in fully developed RPE that lack cilia should be strongly considered/discussed/tested. This concern is highlighted by the fact that the title of the paper starts with "Ablation of primary cilia…" when it should begin with "Knockout of IFT20…".

12. The data described and shown in Figures 7 and 8 are in regard to alleged RPE expressed genes. However, many of them are not genes expressed by the RPE. The majority of the genes shown in Figure 8d are photoreceptor specific genes, including rhodopsin, transducin, recoverin and cone opsin. Unfortunately, this confounds the interpretation of their global gene expression analysis and makes these data have little value. Single cell RNA sequencing would have been a preferred method for this analysis. The fact that the photoreceptor specific genes drastically go up at later time points indicates that more photoreceptor cells were contaminating the RPE cell isolation at these timepoints. How could it be possible to determine what fraction of other gene changes are due to this contamination? Unless this could be addressed, these two figures should be moved to the supplement with a disclaimer that their interpretation is confounded by contamination.

Rev. 2:

This manuscript by Kretschmer and colleagues investigates whether ciliary defects in retinal RPE cells could play a pathological role in retinal dystrophy by creating and characterizing a mouse model that disrupts the primary cilium only in RPE cells. The authors are correct that how primary cilia in the RPE are affecting RPE function and neural retina health has yet to be studied. Their approach to utilize a conditional knockout mouse model to ablate primary cilia exclusively within the RPE cells is very elegant. However, I do find that their conclusions are a bit heavy handed when discussing the impact RPE cilia defects may play in human ciliopathies. While their work does nicely show that ablating the cilia in RPE cells can lead to underlying retinal defects, it should not be presumed that this insult would be greater than the direct insult of loss of ciliary function within the photoreceptor itself. For instance, the authors make a misleading statement in line 28-29 "Our findings that cilia defects in the RPE precede and contribute to retinal degeneration should be considered when designing treatment strategies for retinal degeneration." Here, precede is not accurate as they cannot discount that in humans the ciliary photoreceptor defect would be driving primary retinal dystrophy and any impact from RPE would be secondary. I also have issue with some of the statements and quantifications made regarding TEM images. Finally, I found the organization of the manuscript cumbersome as the final figures showing RPE morphology and transcriptional profiles were underwhelming compared to the earlier figures showing dysfunction of RPE/retina. I would argue that the transcriptional data set and analysis shown in Figure7-8 should be supplemental information as none of the candidates were validated or followed up in any depth. I feel that reorganizing the manuscript would be beneficial.

Major Remarks:

* I am not convinced that the term vacuole is appropriate for what is shown in the TEM images (Figure 4). Vacuoles are defined as fluid-filled spaces surrounded by a membrane that are within a cell. It is not clear that these empty regions are within any cell. They appear to be subretinal gaps between the RPE and photoreceptor outer segments. Considering you already have evidence that there is reduced attachment between RPE and outer segments, I am not surprised that these holes are seen in the TEM images in the Ift20null;Tyrp2-Cre mice, however, referring to them as vacuoles is misleading. It is well documented that fixation artifacts are increased in models of retinal degeneration due to loss of outer segment material and/or photoreceptor cells. There is no evidence that these are cellular vacuoles.

- Additionally, quantification of these vacuoles shown in Figure 4c was not properly performed. The variation between mice must be considered and shown on the graph. The graph must show the #vacuoles counted in each mouse for each genotype. Instead of showing the total # of vacuoles counted.

* The accumulated debris in the Ift20null;Tyrp2-Cre mice TEM image is not well labelled (Figure 4d). The red arrow appears to be pointing at the apical RPE.

* Lines 335-337: "Since our Cre driver is already active during embryonic development, we are not able to address the question of what these remaining ciliated RPE cells are doing and how they might be contributing to RPE function post development."

- Have you examined the Ift20null;Tyrp2-Cre mice to assess % ciliation in adult RPE? If the 12.5% of ciliated RPE cells at E16.5 all end up keeping their cilium in adult, then there may be no difference between controls. The final % ciliation in adult could be important.

* The observations that there are increased macrophages and that RPE cells are growing in area would suggest that the # of RPE cells is reducing. Were RPE cells/area counted?

- More discussion about these results is needed.

Minor Remarks:

* Line 82: I believe this has the wrong citation, should be (14) Jonassen JA et al JCB 2008

* Line 108: dcERG stands for direct-coupled ERGs

* Line 148-149: Is it accurate to say that "At 1-month of age no significant differences could be observed between Ift20null;Tyrp2-Cre mice compared to controls." When in the figure there is a * above the scotopic a-wave at 1 month.

* Figure 3h - the bar graph is missing x-axis labels. Intuitively, I realize each color is time and assume it corresponds to the times collected for the representative images in 3F, but needs to be clear.

* Figure Legend for 3h. "Quantification revealed that the RPE thickness in Ift20null;Tyrp2-Cre mice was significantly lower already at 1-month of age and the subsequent reduction of the retina was accompanied by thinning of the photoreceptor outer segments, while inner segment thickness did not significantly decline until 10-months of age."

- I do not see a graph showing the significant reduction in the RPE layer at 1-month

* Lines 339-341: "What we could show here is that developmental defects in the RPE lead to it a pathological degeneration which precedes retinal degeneration."

- This is a point of discussion but comes across as it was done. Please change the wording to be less predictive and more hypothetical, as it remains a hypothesis. Such as "Using this mouse we could determine whether the developmental defect…"

* Line 691: Typo microilli… should be microvilli

Rev. 3:

The manuscript by Kretschmer et al is a straightforward and elegant study showing that ablation of primary cilia exclusively in the RPE precedes and contribute to photoreceptor degeneration and visual impairment. RPE dysfunction can lead to retinal degeneration (RD) and blindness; the authors now show the role of cilia in RPE in this process. The assays described in the study are robust, and well controlled. I enjoyed reading the manuscript and support immediate publication of the manuscript. Considering my high level of enthusiasm for the clear and important message conveyed by the paper, I mostly have textual and technical comments that might be considered by the authors before final publication.

My major conceptual comment on the role of cilia impinging on RPE dysfunction involves the most intriguing aspect of this work, i.e., the developmental timing of RPE cilia being affected and the delayed effects on RD. The authors mention transcripts related to epithelial homeostasis, the visual cycle and phagocytosis in this process. Especially, EMT in mutated RPEs is discussed as a mechanism for RD. This aspect of the manuscript might be made stronger with further corroboration of the RNASeq data with immunofluorescence and/or immunoblotting of the affected proteins and by including any timeline for such effects.

My major technical comment is regarding data presentation by the authors. Please mention n in all figs, how many experiments or animals tested etc. Show individual data points for counts in graphs. Similarly, please mention SD, SEM, mean/median, box-whisker plots etc in data representation in relevant figures.

Other comments:

1. Lines 79-80: it's unclear how patchy Cre activity was assessed.

2. Lines 86-87: convoluted writing

3. Line 90: Change to ARl13B

4. Lines 130-133: what age photoreceptors and RPE used?

5. Lines 179-182: Authors suggest such vacuoles in RPE not observed before. They do see such structures in control, are these related to the Cre?

6. Lines 198-99: what is meant by depth?

7. Lines 387-390: Not sure if authors' previous paper on Bbs8 ko mentions about EMT changes in the ko RPEs.

8. Please mention methods for melanin quantification in Fig 2.

9. Fig 3h. Please clarify how many controls tested in addition to mutants.

Rev. 4:

The manuscript entitled "Ablation of primary cilia in the RPE leads to retinal degeneration" by Kretschmer and Schneider et al., addresses the intriguing question of what is the role of primary cilia in the RPE cell layer. Although the biogenesis and maintenance of retinal photoreceptor cilia has been investigated in retinal degeneration by numerous studies, the involvement of the cilia made by the RPE layer on retinal degeneration and the underlying disease mechanisms has not been addressed adequately and warrants the current study at hand. Overall the study design and the data presented are of good quality. The authors describe the use of an in-vivo model system to investigate the role of primary cilia of RPE cells utilizing Tyrp2-Cre to delete Ift20. The authors show this leads to retinal degeneration over time and blindness in the conditional KO mice and performs transcriptomic analysis to shed insight into the underlying disease mechanism.

The major caveat of the study however is that it is very descriptive in nature and provides very little mechanistic data on the exact molecular machinery downstream of cilia loss in the RPE leading to retinal degeneration. A revised version of the manuscript addressing the below issues should be considered for publication.

1.) A cilium length quantitation should be added to figure panel 1d.

2.) Figure panel 1F should also show the Tyrp2-Cre only group as a control.

3.) It is difficult to gauge what percentage of RPE cells express Tyrp2-Cre by the images shown in figure panel 1h. This is an important question that needs to be addressed to evaluate if the phenotypes observed are resultant by a partial loss-of function. The number of red cells in comparison to the black RPE layer in the bright field image shown on the right seems quite low.

4.) Figure 2 panel F genotype information is mislabeled and the null group should be identified correctly. There seems to be bleed-through between the DAPI and green channels shown and the authors should utilize arrowheads to identify what was quantified.

5.) The corresponding OCT images for the control group should be shown to each time point instead of the one-month time point as shown in Figure 3f.

6.) Please show equal magnification images of the cut outs in figure 4a. Please show higher magnification images for the Cre only control group and utilize arrows to indicate the vacuoles described.

7.) Can the authors comment on the size of the vacuoles observed and possibly show quantitative data on vacuole size distribution in Figure 4C?

8.) The Bruch's membrane text should not cover the entire membrane in the figure shown hindering its visualization. Maybe utilize BM abbreviation to mark the layer in figure 4e?

9.) Please utilize arrowheads and potentially higher magnification images to point out what is described as "discontinuous Bruch's membrane" in figure 4j.

10.) It is difficult to make any conclusions based on the data shown in figure 6a-c or its significance to the study. Can the authors present the cell size/area data comparing number of cells with different binned sizes at each age? Do the nulls have significantly fewer cells to accommodate a larger area if there is no difference in cell death?

11.) Can the authors provide data on if the retinal vasculature was developed normally in the nulls? Are there any differences in the retinal vasculature over time by the loss of RPE primary cilia?

---

## [Decision Letter · Decision Letter 2]

9 Sep 2023

Dear Dr May-Simera,

Thank you for your patience while we considered your revised manuscript entitled "Ablation of primary cilia, via deletion of IFT20, exclusively in the RPE leads to retinal degeneration" for consideration as a Short Report at PLOS Biology. Your revised study has now been evaluated by the PLOS Biology editors, the Academic Editor and by two of the original reviewers. 

In light of the reviews, which you will find at the end of this email, we are pleased to offer you the opportunity to thoroughly address the remaining points raised by the reviewers in a revision that we anticipate should not take you very long. We will then assess your revised manuscript and your response to the reviewers' comments with our Academic Editor aiming to avoid further rounds of peer-review, although might need to consult with the reviewers, depending on the nature of the revisions.

**IMPORTANT - SUBMITTING YOUR REVISION**

3. Resubmission Checklist

a) *PLOS Data Policy*

You should include the data underlying the graphs shown in the following figures:

Fig. 1B, C, J, K, L; Fig. 2C-G; Fig. 3A-E, G, I; Fig. 4B-E, H; Fig. S1B; Fig. S2B, G; Fig. S3A-D; Fig. S4C and Fig. S5B, D, E

**Please also make publicly available at this stage the data you have deposited in the GEO datadase (GSE144724).

b) *Published Peer Review*

c) *Blurb*

Please also provide a blurb which (if accepted) will be included in our weekly and monthly Electronic Table of Contents, sent out to readers of PLOS Biology, and may be used to promote your article in social media. The blurb should be about 30-40 words long and is subject to editorial changes. It should, without exaggeration, entice people to read your manuscript. It should not be redundant with the title and should not contain acronyms or abbreviations. For examples, view our author guidelines: https://journals.plos.org/plosbiology/s/revising-your-manuscript#loc-blurb

Sincerely,

Ines

--

Ines Alvarez-Garcia, PhD

Senior Editor

PLOS Biology

Reviewers' comments

Rev. 1:

Previous reviewer’s comments:

The manuscript by Kretschmer et al is an interesting study regarding the importance of IFT20 in RPE cells to retinal health. The RPE specific knockout of IFT20, a gene known to be crucial for the maintenance of primary cilia, results in a slow, progressive retinal degeneration associated with defects in RPE cells. The data presented are extensive and mostly of high quality. This study does a nice job in quantification of observed phenotypes. The major fault of the paper is that the phenotype induced by knockout of IFT20 occurs throughout the adult life of the mouse (very slow), when few, if any, RPE cells have a primary cilia. How could the effects of IFT20 knockout be exerted through ciliary defects if RPE do not have cilia to begin with? This point is essentially not addressed. Further, the potential non-ciliary functions of IFT20 are not mentioned or explored. This study should be revised to address these issues and present the data about the phenotype associated with IFT20 deletion from the RPE cells, rather than presume that the effects are through cilia ablation.

AU Response:

We appreciate time and effort that this reviewer took to review our manuscript and are grateful that they acknowledged the quality of the data. Previous reports by ourselves and others had shown that cilia in the RPE are required for development and maturation of the RPE, after which the cilium retracts. In this report we wanted to highlight that this has long lasting consequences, ultimately leading to retinal degeneration. We have now expanded this point in the discussion to make this clearer. We have also updated the discussion to explore the possibility of non-ciliary functions of IFT20. In response to the editor's request, we have thoroughly reorganized the data and figures in this manuscript. We are optimistic that the adoption of the short report format will enhance the conciseness of the story.

- New reviewers’ comments:

The switch to a short report format greatly improves the manuscript by getting the message across more concisely. However, there remains the single most major issue, which has not been addressed adequately in the revision. My principal concern remains that the manuscript still makes the unsubstantiated claim that cilia loss is what causes retinal degeneration, when in fact, this has not been shown. What has been shown, is that loss of IFT20 in the RPE causes retinal degeneration. The most critical place that the authors must correct these claims is in the Title and Abstract.

Previous reviewers’ comments

11. Line 346 - The correct point brought up in Discussion that ciliary proteins may have non-ciliary roles is essentially not addressed in this paper. The disease observed by the authors with IFT20 KO RPE is very slow and occurs over a timeframe when the vast majority of RPE do not have cilia. How could a ciliary defect cause this long-term pathology in cells that lost their cilia during development? Specific non-ciliary roles of IFT20 are not discussed, despite a seemingly abundance of literature on the topic (see PMID: 33072760 for recent review). The concept that IFT20 is doing something important in fully developed RPE that lack cilia should be strongly considered/discussed/tested. This concern is highlighted by the fact that the title of the paper starts with "Ablation of primary cilia…" when it should begin with "Knockout of IFT20…".

AU Response:

We have expanded this point in the discussion and make it clear that non-ciliary functions of IFT20 might also be playing a role here. It is plausible that we are witnessing a compounded effect involving both developmental defects leading to immature/pathogenic RPE and subsequent effects resulting from the loss of IFT20 in adulthood. This effect may arise from the remaining few ciliated cells or from non-ciliary functions of IFT20. To investigate this matter further, we intend to employ inducible IFT20-RPE Cre knockout models in future studies.

- New reviewers’ comments:

I’m happy that you agree that it is still uncertain whether the observed phenotype is due to ciliary loss, or through a different, non-ciliary mechanism. I like your idea to use an inducible IFT20-RPE Cre knockout model to address this question in future studies. However, I am confused why the present manuscript still makes the claim that ciliary loss is causing the effects when, at this point, it is still unknown whether ciliary loss or non-ciliary functions of IFT20 are causing the effects. This unproven claim needs to be corrected in the title, abstract, Figure 4 title, section titles and elsewhere throughout the text.

AU response:

Additionally, we have emphasized that IFT20 remains one of the most appropriate models for investigating the implications of ciliary loss. This is because the loss of other intraflagellar transport proteins (such as IFT88) still allows for the presence of a ciliary stump above the transition zone. Given that IFT20 is essential for ciliary vesicle trafficking to the basal body, its loss generates one of the most comprehensive models for ciliary ablation.

We have also revised the title to reflect the loss of IFT20. However, we find it crucial to emphasize the loss of cilia for readers who may not be familiar with ciliary biology in order to ensure comprehensibility of the article.

- New reviewer’s comments:

The title was changed from:

“Ablation of primary cilia exclusively in the RPE leads to retinal degeneration and visual impairment.”

To:

“Ablation of primary cilia, via deletion of IFT20, exclusively in the RPE leads to retinal degeneration.”

The subject and verb of both sentences does not change, and therefore, both sentences claim “Ablation of primary cilia … leads to retinal degeneration”, which is not shown in this manuscript. A couple examples of appropriate titles would include:

“Deletion of IFT20 exclusively in the RPE leads to retinal degeneration”

“Deletion of IFT20 exclusively in the RPE ablates primary cilia and leads to retinal degeneration”

This same logic in correcting the title, needs to be applied throughout the text and especially in the abstract.

Rev. 2:

The revised manuscript by Kretschmer et al. is better organized and well-written. The authors made major changes to the manuscript in line with the reviewer and editor's comments. I have no major points that need to be addressed and only a few minor comments.

Minor Comments:

DEG is not a commonly used abbreviation and should not be used. Please spell out differentially expressed genes.

A description of the different stages of RPE maturation at P11, P29 and 3-month should be provided early, as these ages are highlighted as being important on line 202-205, but not described.

Line 202-205 - "Although all three time points reflect different stages of RPE maturation and pathogenic processes, it is still interesting to compare differentially expressed transcripts common across all three data points, as these might reflect possible underlying mechanisms (Supplementary Table 4):"

Typo on line 154 - "mayor" should be "major"

Line 277 has a reference formatting error

---

## [Editor Report · Decision Letter 3]

26 Oct 2023

Dear Dr May-Simera,

Thank you for the submission of your revised Short Report entitled "Deletion of IFT20 exclusively in the RPE ablates primary cilia and leads to retinal degeneration." for publication in PLOS Biology. On behalf of my colleagues and the Academic Editor, Dagmar Wachten, I am delighted to let you know that we can in principle accept your manuscript for publication, provided you address any remaining formatting and reporting issues. These will be detailed in an email you should receive within 2-3 business days from our colleagues in the journal operations team; no action is required from you until then. Please note that we will not be able to formally accept your manuscript and schedule it for publication until you have completed any requested changes.

PRESS

Sincerely, 

Ines

--

Ines Alvarez-Garcia, PhD

Senior Editor

PLOS Biology
